# Momentum correlations as signature of sonic Hawking radiation in Bose-Einstein condensates

A. Fabbri[1], N. Pavloff[2]

**1** Centro Fermi – Museo Storico Della Fisica e Centro Studi e Ricerche Enrico Fermi, Piazza del Viminale 1, 00184 Roma, Italy
Dipartimento di Fisica dell'Università di Bologna and INFN Sezione di Bologna, Via Irnerio 46, 40126 Bologna, Italy
Departamento de Física Teórica and IFIC, Universidad de Valencia-CSIC, C. Dr. Moliner 50, 46100 Burjassot, Spain
Laboratoire de Physique Théorique, CNRS UMR 8627, Bât. 210, Univ. Paris-Sud, Université Paris-Saclay, 91405 Orsay Cedex, France
**2** LPTMS, CNRS, Univ. Paris-Sud, Université Paris-Saclay, 91405 Orsay, France

## Abstract

**We study the two-body momentum correlation signal in a quasi one dimensional Bose-Einstein condensate in the presence of a sonic horizon. We identify the relevant correlation lines in momentum space and compute the intensity of the corresponding signal. We consider a set of different experimental procedures and identify the specific issues of each measuring process. We show that some inter-channel correlations, in particular the Hawking quantum-partner one, are particularly well adapted for witnessing quantum non-separability, being resilient to the effects of temperature and/or quantum quenches.**

# 1   Introduction

While the possibility of quantizing gravitation remains elusive, some noticeable progresses have been made in the description of the interaction between the space-time metric and a quantum field. In particular, the dynamical Casimir effect [1, 2] and Hawking radiation from black holes [3] both correspond to a quantum creation of entangled pairs of particles induced by (strong) space-time inhomogeneities, and have both been predicted in the framework of quantum field theory in curved spacetime. In the case of Hawking radiation, the prospect of an experimental study in the genuine astrophysical context seems hopeless, because the radiation has a thermal spectrum at a temperature $T_\mathrm{H} = \hbar\kappa/2\pi c_\ell$, where $c_\ell$ is the speed of light and $\kappa$ the black hole horizon's surface gravity (we use units such that $k_\mathrm{B} = 1$) and in the standard situation of a black hole formed after a gravitational collapse, $T_\mathrm{H}$ is much lower than the temperature of the microwave background radiation [4]. However, the phenomenon of Hawking radiation has a robust kinematic origin, and elaborating on the close analogy of a transonic flow structure with the gravitational metric near a black hole event horizon, Unruh proposed to observe Hawking radiation in a condensed matter context [5]: this idea is often considered as the birth of the field of analogue gravity [6, 7].

Many physical realizations have been proposed for observing analog Hawking radiation (see, e.g., [7–12]), among which the implementation of a sonic horizon in the flow of a Bose-Einstein condensate (BEC) rapidly appeared as quite promising [13]: the low temperature of the system and its paradigmatic quantum nature makes it an ideal playground for studying this phenomenon. However, a direct observation of the analogous sonic radiation in this system is still hindered by thermal effects and difficult to identify unambiguously. The recognition of this difficulty motivated the authors of Refs. [14, 15] to propose the detection of density correlation as an evidence for Hawking emission of correlated pairs of particles from the horizon: Indeed, in an analogous system, contrarily to the gravitational case, the experimentalist is a super-observer who can make measurements from both sides of the horizon. The correlation

between the Hawking particle and its partner were shown in Refs. [14, 15] to induce a characteristic peak in the correlator of density fluctuations which could be used to demonstrate the existence of analogous Hawking radiation in a BEC system. The physical interpretation of this correlated signal is similar to the one initially given by Hawking [3, 16]: Just at the event horizon, vacuum fluctuations produce pairs of virtual quasi-particles, one with negative norm and one with positive norm. The negative norm quasi-particle propagates in the region inside the black hole where it can exist as a real quasi-particle (and is often denoted as the "partner"). The other quasi-particle of the pair is denoted as the "Hawking quantum"; it can escape to infinity, where it constitutes a part of the Hawking radiation. In a BEC the quasi-particles are Bogoliubov excitations which correspond to density fluctuations: hence the emission of the correlated pairs of particles induces density correlations. An interesting aspect of these correlations is that they are resilient to finite temperature effects [17, 18].

In the same line of idea, we proposed in Ref. [19] to study correlations in momentum space as evidence of Hawking radiation. The physical idea is the same as the one behind the study of density correlations in real space, but the specifics are different, with a number of advantages: first, the practical implementation of this type of experiment is well documented [20–23]. Also, it was shown in Ref. [19] that the momentum correlation signal is particularly well adapted to the study of Hawking radiation, being even less affected by the background temperature than the real-space correlation signal, and offers a clear and robust signature of the entangled nature of the Hawking pairs. In the present paper we develop and explicit the results presented in Ref. [19]. We detail the theoretical description of the quantum fluctuation of the system and precise how a local Fourier analysis can be performed. This leads us to underline some characteristics of the experimental detection scheme which are crucial for the detection of entanglement (cf. the discussions in Appendix A and at the end of Sec. 3.1). We also extend the treatment of Ref. [19] in order to include what we denote as "non adiabatic effects" and "*in situ* measurements" (Sec. 3.3). We show that the results presented in Ref. [19] are robust with respect to this more general treatment, and that new correlation lines appear which, at variance with the previous ones, show no signature of non-separability.

Another important motivation of our work is the recent experimental study of Steinhauer [24] who studied an acoustic BEC black hole in one of the models discussed below (the so called "waterfall model" [25]) and presented results on entanglement similar to the ones discussed below.

The paper is organized as follows: Sec. 2 presents several black hole configurations in a quasi one dimensional BEC system. In Sec. 3 we compute the corresponding theoretical momentum correlation functions and the non-separability signals in the different configurations in a variety of situations: In particular we present the adiabatic and non-adiabatic regimes and also address in both cases the effects of temperature. These results are compared in Sec. 4 with the ones obtained in the absence of sonic horizon. In section 5 we discuss the limitations of our theoretical approach and finally we present our conclusion in section 6. Some technical points are presented in the Appendices: in Appendix A we discuss a rigorous windowed Fourier analysis which induces important restrictions to the measurement process; in Appendix B we give the form of the most general correlation functions and in Appendix C we discuss the specific case of a subsonic flow in the presence of a localized external potential.

# 2 Black hole configurations and their theoretical description

## 2.1 Quasi one-dimensional sonic black holes

In this work we consider a system where bosons are confined in one dimension by a harmonic transverse potential of angular frequency $\omega_\perp$. We denote by $x$ the longitudinal degree of freedom and assume no trapping along $x$. In this configuration the theoretical description of the system is conveniently worked out in the quasi one-dimensional limit where the particles are described by a one dimensional (1D) quantum field $\hat{\Psi}(x)$. According to the Bogoliubov prescription one writes the field operator as the sum of a main contribution (a classical field $\Psi_0$) and a small quantum remnant

$$\hat{\Psi}(x) = \Psi_0(x) + \hat{\psi}(x) \; . \tag{1}$$

$\Psi_0(x)$ describes the condensate order parameter and verifies the stationary 1D Gross-Pitaevskii equation

$$\mu \, \Psi_0 = -\frac{\hbar^2}{2\,m}\partial_x^2\Psi_0 + \left[ U(x) + g_{1d}|\Psi_0|^2 \right] \Psi_0 \; , \tag{2}$$

with $g_{1d} = 2\hbar\omega_\perp a$ [26], where $a$ is the 3D s-wave scattering length. In (2) $\mu$ is the chemical potential and $U(x)$ a possible longitudinal external potential. In the absence of external potential, for a static homogeneous system of constant linear density $|\Psi_0|^2 = n$ one gets $\mu = g_{1d}n$. Useful quantities are the sound velocity in the uniform system $c = \sqrt{\mu/m}$ and the healing length $\xi = \hbar/mc$.

In 1D a description based on Equations (1) and (2) is not quite legitimate, both in the high and in the low density limit: at large density, transverse excitations of the condensate cannot be discarded and the quasi 1D description (2) fails; at low density, phase fluctuations destroy the long range order and the possibility of a true Bose-Einstein condensation which is at the heart of the Bogoliubov description (1). In the remaining of this section we stick to the simple approach embodied by Equations (1) and (2) and we postpone the discussion of its limitations to Sec. 5.

We denote as a black hole configuration a 1D configuration in which the asymptotic upstream flow is subsonic (with constant density $n_u$) and the asymptotic downstream one is supersonic (with constant density $n_d$). Typically $n_u \neq n_d$ and when a region of the flow is denoted for instance as subsonic, this means that in this region the density of the condensate is constant, and its velocity $V_u$ is smaller than the asymptotic sound velocity $c_u = \sqrt{g_{1d}n_u/m}$.

Several black hole configurations have been proposed in Refs. [14, 15, 17]. The specific form of the order parameter is always of the type:

$$\Psi_0(x) = \begin{cases} \sqrt{n_u}\exp(\mathrm{i}K_u x)\,\phi_u(x) & \text{for} \quad x < 0, \\ \sqrt{n_d}\exp(\mathrm{i}K_d x)\,\phi_d(x) & \text{for} \quad x > 0, \end{cases} \tag{3}$$

where $K_{u,d} = mV_{u,d}/\hbar$, $V_u$ being the asymptotic upstream flow velocity and $V_d$ the downstream one ($V_u$ and $V_d$ are both positive). We also introduce the healing lengths $\xi_\alpha = \hbar/(mc_\alpha)$ and the Mach numbers $M_\alpha = V_\alpha/c_\alpha$ ($\alpha = u$ or $d$ depending if one considers upstream or downstream quantities). The functions $\phi_u$ and $\phi_d$ verify $|\phi_d(x)| = 1$ and $\lim_{x\to-\infty}|\phi_u(x)| = 1$. The asymptotic upstream and downstream flows are respectively subsonic and supersonic, meaning that $M_u < 1 < M_d$. A remark on the location of the event horizon is in order here. First, as in any analogous system, its actual position is energy-dependent: we will even see

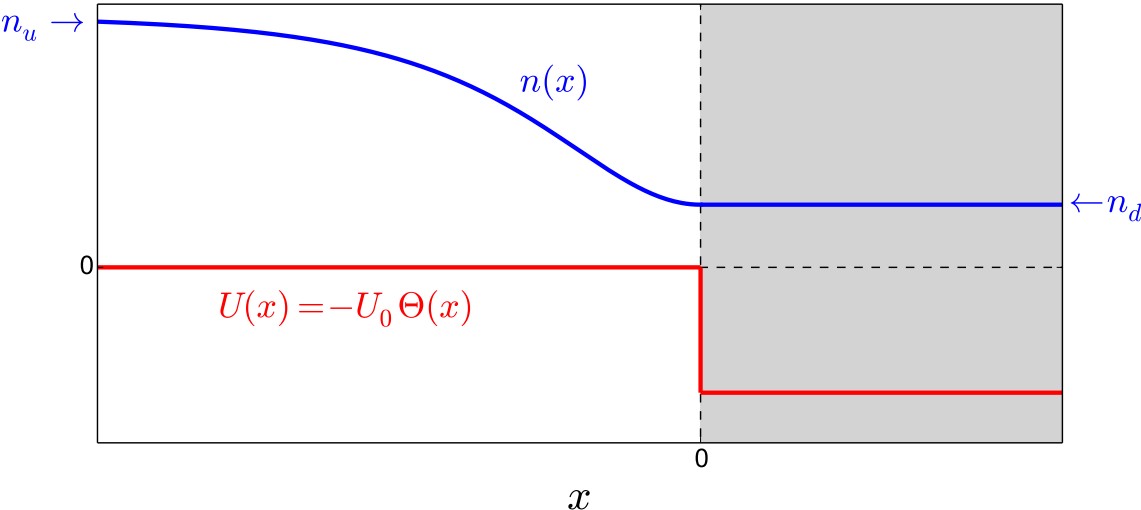

Figure 1: Waterfall configuration. The flow is incident from the left with an asymptotic density $n_u$ and a (subsonic) velocity $V_u$. The downstream ($x > 0$) velocity $V_d$ is supersonic. The downstream density $n_d$ is constant and lower than $n_u$. The region $x > 0$ is shaded in order to recall that it corresponds to the interior of the black hole.

below that the horizon disappears at large energy. This effect being taken into account, the customary procedure is to do a semi-classical analysis and to define as the true horizon the large wavelength one. In this case the horizon is the point where the flow velocity is equal to the local speed of sound. However, the flow varies rapidly around $x = 0$ in the configurations we study below, and a quantity such as the local speed of sound is ill defined in this region. As a result, the position of the event horizon cannot be unambiguously defined. This is not a drawback of the model: what really matters is that the asymptotic upstream and downstream flows are truly respectively sub- and super-sonic.

### 2.1.1 The "waterfall" configuration

We first consider one of the realistic configurations introduced in Ref. [25] and realized experimentally in Ref. [24]. In this configuration, which we denote as "waterfall", the 1D flow of a BEC is subject to an external potential which is a step function of the form $U(x) = -U_0 \, \Theta(x)$, where $\Theta$ is the Heaviside function (and $U_0 > 0$). In this case, a stationary profile with a flow which is subsonic upstream and supersonic downstream, i.e., a black hole configuration, has been identified in Refs. [24, 27] and is schematically represented in Fig. 1. The upstream profile is exactly one half of a dark soliton and the downstream one corresponds to a flow with constant density and velocity (cf. Fig. 1). In the waterfall configuration one has $\phi_d(x) = -\mathrm{i}$ and $\phi_u(x) = \cos\theta \tanh(x\cos\theta/\xi_u) - \mathrm{i}\sin\theta$, where $\sin\theta = M_u$. One has also $U_0/g_{1\mathrm{d}}n_u = \frac{1}{2}(M_u^2 + M_u^{-2}) - 1$ and $V_u = c_d < c_u < V_d$ which indeed corresponds to a black hole type of horizon ($M_u < 1 < M_d$).

### 2.1.2   The "$\delta$ peak" configuration

In this configuration the 1D flow of a BEC is subject to an external potential which is a Dirac distribution of the form $U(x) = \kappa \, \delta(x)$, where $\kappa > 0$. In this case, a stationary profile with a flow which is subsonic upstream and supersonic downstream, i.e., a black hole configuration, has been identified in Refs. [28, 29], and it has been shown in Ref. [30] how this configuration can be reached dynamically. The downstream one corresponds to a flow with constant density and velocity and the upstream profile is a fraction of a dark soliton, with $\phi_u(x) = \cos\theta \tanh[(x - x_0)\cos\theta/\xi_u] - \mathrm{i}\sin\theta$, where $\sin\theta = M_u$ and $x_0$ depends on $M_u$ and $\kappa$ (see details in [25]).

### 2.1.3   The "flat profile" configuration

We finally present a model configuration first introduced in Ref. [15], which has been denoted as "flat profile" in Ref. [25]. Although this configuration is not likely to be realized experimentally, it has been demonstrated in Ref. [25] that it yields a density correlation signal which is quite similar to the one obtained in the more realistic waterfall and $\delta-$peak configurations. We will use the flat profile configuration to present our results below (in Sec. 3) for pedagogical reasons, because it leads to a simpler phenomenology for the momentum correlation than the other configurations.

In the flat profile configuration one has $n_u = n_d \equiv n_0$ and $K_u = K_d \equiv K_0$ and the $\phi_\alpha$ functions of Eq. (3) assume a very simple value: $\phi_u(x) = \phi_d(x) = 1$. This means that $\Psi_0(x)$ is a plane wave for all $x$. A horizon can still be realized in this case by tuning the values of the external potential $U(x)$ and of the non-linear constant $g_{1\mathrm{d}}(x)$ such that

$$U(x) = \begin{cases} U_u & \text{for} \quad x < 0, \\ U_d & \text{for} \quad x > 0, \end{cases} \quad \text{and} \quad g_{1\mathrm{d}}(x) = \begin{cases} g_u & \text{for} \quad x < 0, \\ g_d & \text{for} \quad x > 0. \end{cases} \tag{4}$$

These values are chosen so that a flow with $\Psi_0(x) = \sqrt{n_0}\exp(\mathrm{i}K_0 x)$ is solution of Eq. (2) for all $x$. This imposes

$$\frac{c_d}{c_u} = \frac{M_u}{M_d} = \frac{\xi_u}{\xi_d}\,, \quad \text{and} \quad g_u n_0 + U_u = g_d n_0 + U_d\,. \tag{5}$$

We finally note that in the flat profile configuration one has $c_d < V_d = V_u < c_u$. This corresponds to a sonic black hole horizon since the upstream and downstream Mach numbers verify $M_u < 1 < M_d$.

It is important to notice that, at variance with the cases of the waterfall and of the $\delta$ peak configurations, where, once an asymptotic flow is fixed (say, the upstream one) all the characteristics of the flow are uniquely determined, in the case of the flat profile configuration the values of $M_u$ and $M_d$ can be chosen independently one from the other. As a result, one cannot directly compare the results of, say the measure of non separability, of a waterfall and a $\delta$ peak configuration, but each of them can be compared with a flat profile configuration. This will be done in Figs. 4, 5, 6 and 7.

## 2.2   The excitation spectrum of a homogeneous condensate

In the case of a static homogeneous condensate, the dispersion relation of longitudinal excitations is the standard Bogoliubov one:

$$\omega = \omega_{\mathrm{B}}(q) = c\,|q|\left(1 + \tfrac{1}{4}\xi^2 q^2\right)^{1/2}\,. \tag{6}$$

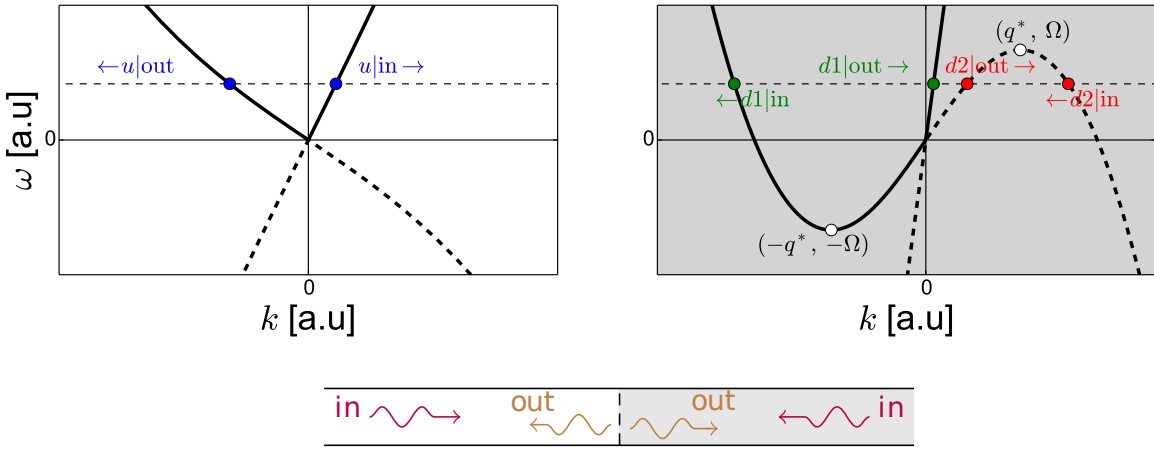

Figure 2: Dispersion relations. The left plot corresponds to a subsonic flow. The right plot corresponds to a supersonic flow; it is shaded in order to recall that it describes the situation inside the black hole. In each plot the horizontal dashed line is fixed by the chosen value of $\omega$. The $q_\ell(\omega)$'s are the corresponding abscissas, with $\ell \in \{u|\text{in}, u|\text{out}\}$ in the left plot and $\ell \in \{d1|\text{in}, d1|\text{out}, d2|\text{in}, d2|\text{out}\}$ in the right plot. The direction of propagation (left or right) of the eigen-modes is represented by an arrow. The lower diagram illustrates schematically the "ingoing" and "outgoing" terminology used in the text and in the two upper plots.

In a region where the condensate flows with a velocity $V$ this is modified to

$$(\omega - Vq)^2 = \omega_{\text{B}}^2(q) \ . \tag{7}$$

In this case $\omega$ is the energy of the elementary excitation in the frame where the obstacle is at rest, while $\omega_{\text{B}}$ is the frequency measured in the frame of the fluid. The momentum of the excitation relative to the background flow is $\hbar q$, and its momentum in the frame of the obstacle is $\hbar q + mV$. In a black hole configuration, the upstream and the downstream channels are characterized by dispersion relations of the type (7) (with the appropriate values of $V$, $\xi$ and $c$), they are illustrated in Fig. 2. In this figure the upper left (upper right) plot represents the asymptotic upstream subsonic (downstream supersonic) case. The part of the dispersion relation represented by a dashed line correspond to negative norm modes, as explained in the following section. In all this work we only consider the $\omega > 0$ part of the dispersion relations, this is made possible by the $\omega \leftrightarrow -\omega$ symmetry of the spectrum [31].

One sees in the figure that, upstream, the waves of one of the channels are directed towards the horizon (ingoing waves, denoted as $u|\text{in}$) whereas the waves in the other channel propagate away from the horizon (outgoing waves: $u|\text{out}$). The definition of which mode is ingoing and which is outgoing depends on the side of the horizon that is considered, this is illustrated by the lower diagram of Fig. 2. Downstream, there are two ingoing waves ($d1|\text{in}$ and $d2|\text{in}$) and two outgoing waves ($d1|\text{out}$ and $d2|\text{out}$). Note that the two $d2$ channels disappear at large energy, when $\omega > \Omega$, see Fig. 2. $\Omega$ is the energy of an elementary excitation whose group velocity in the frame where the condensate is at rest ($\partial \omega_{\text{B}} / \partial q$) is equal to the flow velocity $V_d$ (such an equality is only possible for supersonic flows). Excitations with momentum larger than the one of this excitation (which is denoted as $q^*$ in Fig. 2) move faster than the flow and are able to escape the "black hole".

## 2.3 The wave function in real space

The field operator $\hat{\psi}(x)$ associated in the Schrödinger representation to the particles which are out of the condensate [as defined by Eq. (1)] is expanded over the scattering modes:

$$
\begin{aligned}
\hat{\psi}(x) &= \mathrm{e}^{\mathrm{i}K_\alpha x} \int_0^\infty \frac{\mathrm{d}\omega}{\sqrt{2\pi}} \sum_{L\in\{U,D1\}} \left[ \bar{u}_L(x,\omega)\hat{b}_L(\omega) + \bar{w}_L^*(x,\omega)\hat{b}_L^\dagger(\omega) \right] \\
&+ \mathrm{e}^{\mathrm{i}K_\alpha x} \int_0^\Omega \frac{\mathrm{d}\omega}{\sqrt{2\pi}} \left[ \bar{u}_{D2}(x,\omega)\hat{b}_{D2}^\dagger(\omega) + \bar{w}_{D2}^*(x,\omega)\hat{b}_{D2}(\omega) \right] .
\end{aligned}
\tag{8}
$$

$K_\alpha$ in (8) is equal to $K_u$ if $x < 0$ and to $K_d$ if $x > 0$. The $\hat{b}_L^\dagger(\omega)$'s create an excitation of energy $\hbar\omega$ in one of the three scattering modes ($L = U$, $D1$ or $D2$), they obey the following commutation relation:

$$
[\hat{b}_L(\omega), \hat{b}_{L'}^\dagger(\omega')] = \delta_{L,L'}\delta(\omega - \omega').
\tag{9}
$$

Each of the three scattering modes ($U$, $D1$ or $D2$) is initiated by one of the three entrance channels ($u|$in, $d1|$in or $d2|$in). Far from the horizon, the density and the velocity of the flow are position-independent and the corresponding wave functions are mere plane waves of the following form:

- Deep in the upstream subsonic region, i.e., for $x < 0$, $x \ll -\xi_u$ :

$$
\begin{aligned}
\begin{pmatrix} \bar{u}_U(x) \\ \bar{w}_U(x) \end{pmatrix} &= S_{u,u} \begin{pmatrix} \mathcal{U}_{u|\text{out}} \\ \mathcal{W}_{u|\text{out}} \end{pmatrix} \mathrm{e}^{\mathrm{i}q_{u|\text{out}}x} + \begin{pmatrix} \mathcal{U}_{u|\text{in}} \\ \mathcal{W}_{u|\text{in}} \end{pmatrix} \mathrm{e}^{\mathrm{i}q_{u|\text{in}}x} , \\
\begin{pmatrix} \bar{u}_{D1}(x) \\ \bar{w}_{D1}(x) \end{pmatrix} &= S_{u,d1} \begin{pmatrix} \mathcal{U}_{u|\text{out}} \\ \mathcal{W}_{u|\text{out}} \end{pmatrix} \mathrm{e}^{\mathrm{i}q_{u|\text{out}}x} , \\
\begin{pmatrix} \bar{u}_{D2}(x) \\ \bar{w}_{D2}(x) \end{pmatrix} &= S_{u,d2} \begin{pmatrix} \mathcal{U}_{u|\text{out}} \\ \mathcal{W}_{u|\text{out}} \end{pmatrix} \mathrm{e}^{\mathrm{i}q_{u|\text{out}}x} .
\end{aligned}
\tag{10}
$$

- Deep in the downstream supersonic region, i.e., when $x > 0$, $x \gg \xi_d$ :

$$
\begin{aligned}
\begin{pmatrix} \bar{u}_U(x) \\ \bar{w}_U(x) \end{pmatrix} &= S_{d1,u} \begin{pmatrix} \mathcal{U}_{d1|\text{out}} \\ \mathcal{W}_{d1|\text{out}} \end{pmatrix} \mathrm{e}^{\mathrm{i}q_{d1|\text{out}}x} + S_{d2,u} \begin{pmatrix} \mathcal{U}_{d2|\text{out}} \\ \mathcal{W}_{d2|\text{out}} \end{pmatrix} \mathrm{e}^{\mathrm{i}q_{d2|\text{out}}x} , \\
\begin{pmatrix} \bar{u}_{D1}(x) \\ \bar{w}_{D1}(x) \end{pmatrix} &= S_{d1,d1} \begin{pmatrix} \mathcal{U}_{d1|\text{out}} \\ \mathcal{W}_{d1|\text{out}} \end{pmatrix} \mathrm{e}^{\mathrm{i}q_{d1|\text{out}}x} + S_{d2,d1} \begin{pmatrix} \mathcal{U}_{d2|\text{out}} \\ \mathcal{W}_{d2|\text{out}} \end{pmatrix} \mathrm{e}^{\mathrm{i}q_{d2|\text{out}}x} + \begin{pmatrix} \mathcal{U}_{d1|\text{in}} \\ \mathcal{W}_{d1|\text{in}} \end{pmatrix} \mathrm{e}^{\mathrm{i}q_{d1|\text{in}}x} , \\
\begin{pmatrix} \bar{u}_{D2}(x) \\ \bar{w}_{D2}(x) \end{pmatrix} &= S_{d1,d2} \begin{pmatrix} \mathcal{U}_{d1|\text{out}} \\ \mathcal{W}_{d1|\text{out}} \end{pmatrix} \mathrm{e}^{\mathrm{i}q_{d1|\text{out}}x} + S_{d2,d2} \begin{pmatrix} \mathcal{U}_{d2|\text{out}} \\ \mathcal{W}_{d2|\text{out}} \end{pmatrix} \mathrm{e}^{\mathrm{i}q_{d2|\text{out}}x} + \begin{pmatrix} \mathcal{U}_{d2|\text{in}} \\ \mathcal{W}_{d2|\text{in}} \end{pmatrix} \mathrm{e}^{\mathrm{i}q_{d2|\text{in}}x} .
\end{aligned}
\tag{11}
$$

Note that the $\bar{u}_L$'s, $\bar{w}_L$'s, $q_\ell$'s, $S_{i,j}$'s, $\mathcal{U}_\ell$'s and the $\mathcal{W}_\ell$'s in Eqs. (10) and (11) all depend on $\omega$. For instance $q_\ell(\omega)$ is defined on Fig. 2 for $\ell \in \{u|\text{out}, u|\text{in}, d1|\text{out}, d1|\text{in}, d2|\text{out}, d2|\text{in}\}$. We chose a normalization of the the coefficients $\mathcal{U}_\ell$ and $\mathcal{W}_\ell$ – the so called "Bogoliubov amplitudes" – such that

$$
|\mathcal{U}_\ell(\omega)|^2 - |\mathcal{W}_\ell(\omega)|^2 = \frac{\pm 1}{|\partial\omega/\partial q_\ell|}.
\tag{12}
$$

The sign $+$ $(-)$ in (12) refers to positive (negative) norm modes. All the upstream modes ($u|$in and $u|$out) have a positive norm. In the downstream region, the $d1|$in and $d1|$out modes have a positive norm while the $d2|$in and $d2|$out ones have a negative norm. The normalization (12) ensures that a positive (negative) mode carries a current $+1$ $(-1)$. The explicit expression

of the the coefficients $\mathcal{U}_\ell(\omega)$ and $\mathcal{W}_\ell(\omega)$ corresponding to the normalization (12) is given in Ref. [25]. Expressions (10) are not valid close to the horizon due to (i) the modification of the density profile which is position-dependent in vicinity of the horizon[1] and (ii) to the occurrence of evanescent modes (with complex momenta solutions of Eq. (7)) which are of importance near the horizon.

Let us for instance discuss the physical content of the last of Eqs. (11). It describes a $d2|\text{in}$ wave incoming from $+\infty$ (last term of the r.h.s., the corresponding group velocity is negative, cf. Fig. 2) which is back scattered into a $d1|\text{out}$ and a $d2|\text{out}$ wave, with respective reflection amplitudes $S_{d1,d2}$ and $S_{d2,d2}$. Part of this wave is also transmitted in the $x < 0$ region as a $u|\text{out}$ wave with transmission amplitude $S_{u,d2}$ : the corresponding expression far from the horizon is displayed in the last of Eqs. (10). The scattering amplitudes form the $S$ matrix which is $3 \times 3$ for energies $\omega$ lower than the threshold $\Omega$ defined on Fig. 2:

$$
S = \begin{pmatrix} S_{u,u} & S_{u,d1} & S_{u,d2} \\ S_{d1,u} & S_{d1,d1} & S_{d1,d2} \\ S_{d2,u} & S_{d2,d1} & S_{d2,d2} \end{pmatrix} . \tag{13}
$$

Current conservation reads

$$
S^\dagger \eta S = \eta = S \eta S^\dagger, \quad \text{where} \quad \eta = \mathrm{diag}(1, 1, -1). \tag{14}
$$

For $\omega > \Omega$, the last row and the last column of (13) vanish because the $d2$ mode disappears. In this case the $S$ matrix is $2 \times 2$ and satisfies $SS^\dagger = \mathbb{1}$.

In Eqs. (10) and (11) we did not write the contribution of the evanescent modes since they decay exponentially and are negligible far from the horizon (when $|x| \gg \xi_{(u,d)}$), but we fully take these modes into account in the expression of the $\bar{u}$'s and the $\bar{w}$'s near the horizon (around $x = 0$); this is needed for an accurate computation of the $S$ matrix. In a given configuration (say "waterfall"), the elements of the $S$ matrix are determined for each value of $\omega$ by enforcing continuity of the wave functions (and of their spatial derivatives) of the elementary excitations at $x = 0$. This represents an easy numerical task which consists in solving a linear $4 \times 4$ system for each value of $\omega$.

## 2.4 The wave function in momentum space

Because of the presence of the negative norm/negative energy $d2|\text{in}$ mode, stationary black hole configurations such as the ones presented in section 2.1 are meta-stable[2]. Indeed, when reaching the horizon, the $d2|\text{in}$ waves give rise to a radiation of $u|\text{out}$ quanta in the upstream region, which constitutes the spontaneous Hawking radiation. In BEC systems however, this radiation is not easily detected. The reason is that the occupation of the Hawking radiating modes is approximately of thermal type, with an effective temperature much lower than the true temperature of the system (typically by a factor 10), and the Hawking signal is thus drowned in the thermal noise. This circumstance led the authors of Refs. [14, 15] to look for density correlations as alternative evidence of the Hawking effect.

---

[1]Note however that the density is not affected by the horizon in the flat profile configuration. In the waterfall and delta peak configurations, the corresponding explicit form, correct even close to the horizon, is given in Ref. [25].

[2]In the gravitational context the $d2|\text{in}$ mode is absent, and this metastability arises when the black hole is dynamically created.

The idea, checked in Refs. [14, 15] is that outgoing waves generated by the same $d2|$in mode are all correlated. Moreover, since the corresponding amplitudes are governed by the $S$ matrix which describes how ingoing waves hitting the horizon generate outgoing ones, the knowledge of the $S$ matrix makes a detailed description of the correlation signal possible. This point has been checked thoroughly in Refs. [17, 18, 25]. In particular, the characteristic peaks of the density correlations correspond to the Hawking quantum ($u|$out) - partner ($d2|$out) correlations (for points situated on both sides of the horizon), and also to correlations along the $u|$out$- d1|$out (again, for points situated on both sides of the horizon) and $d2|$out$- d1|$out (for both points inside the horizon) channels.

In a BEC, momentum correlations could be more precisely detected than density correlations, by following a procedure used in Ref. [22] in a similar context, in the case of the dynamical Casimir effect. This is the reason why we proposed in Ref. [19] to demonstrate the existence of sonic Hawking radiation by the means of correlations in momentum space.

### 2.4.1 A local Fourier transform

By appropriately introducing a local Fourier transform in both the subsonic (exterior of the black hole) and supersonic (black hole interior) regions, we shall explicitly construct the momentum correlator and analyze, in Sec. 3, its nontrivial qualitative features, which are in correspondence with those present in the density correlation signal. This signal concerns the occupation number in the momentum representation: $\hat{N}(K) = \hat{\psi}^\dagger(K)\hat{\psi}(K)$, where $\hat{\psi}(K)$ is the Fourier transform of $\hat{\psi}(x)^3$:

$$\hat{\psi}(K) = \frac{1}{\sqrt{2\pi}} \int_{\mathbb{R}} \mathrm{d}x \exp\{-\mathrm{i}Kx\}\hat{\psi}(x) . \qquad (15)$$

From expression (8) and the mode analysis presented in Secs. 2.2 and 2.3, it is clear that the momentum distribution has a different form in the far-upstream and far-downstream regions. Hence, instead of (15), it is more appropriate to perform a specific mode analysis in each of these regions [32–34]. This can be done by using a window function selecting the desired region of space. The precise form of this window function is irrelevant, but for concreteness we will consider a Gaussian. In the upstream region for instance, one takes a window

$$\Pi_u(x) = \Lambda_u \exp\left\{-\frac{(x - X_u)^2}{\sigma_u^2}\right\} , \qquad (16)$$

and the corresponding windowed Fourier transform is

$$\hat{\psi}_u(K) = \frac{1}{\sqrt{2\pi}} \int_{\mathbb{R}} \mathrm{d}x \exp\{-\mathrm{i}Kx\}\Pi_u(x)\hat{\psi}(x) . \qquad (17)$$

This procedure is meant to select the momentum components which can be identified from (10). For this purpose, the parameters of the window function have to be chosen in order to work in the appropriate region of space. This is done by taking $X_u < 0$, $\sigma_u > 0$ and letting $X_u$ and $\sigma_u$ respectively tend to $-\infty$ and $+\infty$, imposing for the ratio $X_u/\sigma_u \to -1$ which allows that $X_u + \sigma_u = C^{\mathrm{st}} \ll -\xi_u$. This procedure is illustrated in Fig. 3. It is important to take the limit of large $\sigma_u$ and $|X_u|$ for obtaining a sharp ($\delta$-like) distribution in momentum

---

[3]We only consider here the momentum distribution of particles which are outside of the condensate and discard the contribution of the condensate.

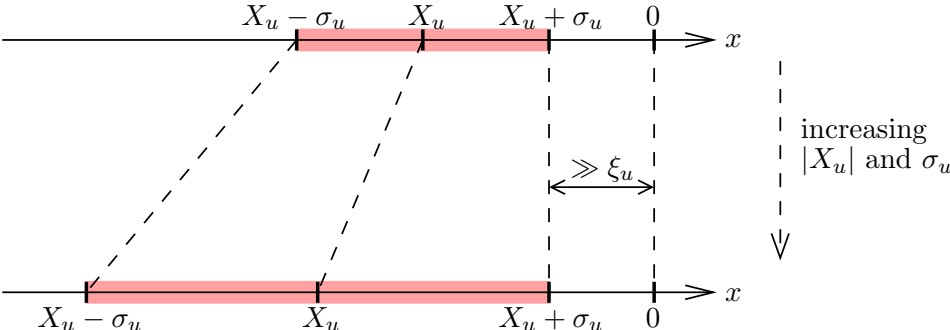

Figure 3: Schematic representation of the behavior of the parameters $X_u$ and $\sigma_u$ of the window function $\Pi_u(x)$ defined in Eq. (16). The shaded zone is the region where the window function notably differs from zero. The spacing between $X_u + \sigma_u$ and the origin is large compared to $\xi_u$, and this ensures that the Fourier analysis, which is made local thanks to the window function, is performed in the deep subsonic region.

space. In Eq. (16) the normalization parameter $\Lambda_u$ is introduced to effectively describe the finite efficiency of the experimental detection apparatus. More precisely, it is argued in Sec. 4 that the quantity $\lambda_u = \sigma_u \Lambda_u^2 / \sqrt{2\pi}$ describes the rate of detection of particles in the window $[X_u - \sigma_u, X_u + \sigma_u]$: if $\lambda_u = 0$ none of the particles are detected, if instead $\lambda_u = 1$, they are all detected. It is interesting to stress that the explicit results given in Sec. 3 for the normalized momentum correlation function (29) do not depend on the efficiencies $\lambda_u$ and $\lambda_d$ of the detectors.

Of course, a local Fourier transform similar to (17) is performed downstream, using a different window $\Pi_d(x)$ with parameters $\Lambda_d$, $X_d$ ($> 0$) and $\sigma_d$, leading to the downstream Fourier transform $\hat{\psi}_d(K)$. We will see in Appendix A that the parameters of the upstream and of the downstream window function have to be chosen in a specific manner when one wants to consider a specific correlation signal. However, these precise conditions – which are of importance for the theoretical analysis of the experimental detection scheme – can be replaced, once met, by the following schematic, but natural rules:

R1 : A contribution to (8), which, for $x < 0$, reads $e^{iK_u x} \int d\omega\, C^{st} e^{\pm iq_\ell(\omega)x}$ yields a contribution $\sqrt{2\pi} \int d\omega\, C^{st} \delta(K - (K_u \pm q_\ell(\omega)))$ to the upstream Fourier transform $\hat{\psi}_u(K)$.

R2 : A contribution to (8), which, for $x > 0$, reads $e^{iK_d x} \int d\omega\, C^{st} e^{\pm iq_\ell(\omega)x}$ yields a contribution $\sqrt{2\pi} \int d\omega\, C^{st} \delta(K - (K_d \pm q_\ell(\omega)))$ to the downstream Fourier transform $\hat{\psi}_d(K)$.

These rules are less rigorous than the correct mathematical procedure (17) which uses window functions for defining the local Fourier transforms, but it is shown in Appendix A that, provided some simple and natural redefinitions (in particular of the singular Dirac distributions) are considered, they yield the correct result. Hence, we shift discussion of the more rigorous results to Appendix A and present our results in the main text using these simplified rules. Performing the schematic Fourier transform in the $x < 0$ region we get,

$$\hat{\psi}_u(K) = \int_0^\infty d\omega \quad \left\{ \quad \delta(K - K_u - q_{u|\text{out}}) \mathcal{U}_{u|\text{out}}(S_{u,u}\hat{b}_U + S_{u,d1}\hat{b}_{D1} + S_{u,d2}\hat{b}_{D2}^\dagger) \right.$$

$$+ \quad \delta(K - K_u + q_{u|\text{out}})\mathcal{W}^*_{u|\text{out}}(S^*_{u,u}\hat{b}^\dagger_U + S^*_{u,d1}\hat{b}^\dagger_{D1} + S^*_{u,d2}\hat{b}_{D2})$$
$$+ \quad \delta(K - K_u + q_{u|\text{in}})\mathcal{W}^*_{u|\text{in}}\hat{b}^\dagger_U$$
$$+ \quad \delta(K - K_u - q_{u|\text{in}})\mathcal{U}_{u|\text{in}}\hat{b}_U \Big\} \,, \tag{18}$$

whereas in the $x > 0$ one gets

$$\hat{\psi}_d(K) = \int_0^\infty \mathrm{d}\omega \quad \Big\{ \quad \delta(K - K_d - q_{d1|\text{out}})\mathcal{U}_{d1|\text{out}}(S_{d1,u}\hat{b}_U + S_{d1,d1}\hat{b}_{D1} + S_{d1,d2}\hat{b}^\dagger_{D2})$$
$$+ \quad \delta(K - K_d + q_{d1|\text{out}})\mathcal{W}^*_{d1|\text{out}}(S^*_{d1,u}\hat{b}^\dagger_U + S^*_{d1,d1}\hat{b}^\dagger_{D1} + S^*_{d1,d2}\hat{b}_{D2})$$
$$+ \quad \delta(K - K_d - q_{d2|\text{out}})\mathcal{U}_{d2|\text{out}}(S_{d2,u}\hat{b}_U + S_{d2,d1}\hat{b}_{D1} + S_{d2,d2}\hat{b}^\dagger_{D2})$$
$$+ \quad \delta(K - K_d + q_{d2|\text{out}})\mathcal{W}^*_{d2|\text{out}}(S^*_{d2,u}\hat{b}^\dagger_U + S^*_{d2,d1}\hat{b}^\dagger_{D1} + S^*_{d2,d2}\hat{b}_{D2})$$
$$+ \quad \delta(K - K_d + q_{d1|\text{in}})\mathcal{W}^*_{d1|\text{in}}\hat{b}^\dagger_{D1}$$
$$+ \quad \delta(K - K_d - q_{d1|\text{in}})\mathcal{U}_{d1|\text{in}}\hat{b}_{D1}$$
$$+ \quad \delta(K - K_d - q_{d2|\text{in}})\mathcal{U}_{d2|\text{in}}\hat{b}^\dagger_{D2}$$
$$+ \quad \delta(K - K_d + q_{d2|\text{in}})\mathcal{W}^*_{d2|\text{in}}\hat{b}_{D2} \Big\} \,. \tag{19}$$

In the two above integrals the $q_\ell$'s are functions of $\omega$ computed as schematically represented in Fig. 2. The $\omega$ integration yields factors $|\partial q_\ell / \partial \omega|$ which can be absorbed in a re-definition of the $\mathcal{U}$'s and of the $\mathcal{W}$'s:

$$\widetilde{\mathcal{U}}_\ell(q) = \mathcal{U}_\ell(\omega_\ell(q)) \left|\frac{\partial \omega_\ell}{\partial q}\right| \text{ and } \widetilde{\mathcal{W}}_\ell(q) = \mathcal{W}_\ell(\omega_\ell(-q)) \left|\frac{\partial \omega_\ell}{\partial q}\right| \,, \tag{20}$$

where $\omega_\ell(q)$ is the reciprocal function of $q_\ell(\omega)$. The "tilde Bogoliubov coefficients" satisfy the following normalization:

$$|\widetilde{\mathcal{U}}_\ell(q)|^2 - |\widetilde{\mathcal{W}}_\ell(q)|^2 = \pm \left|\frac{\partial \omega_\ell}{\partial q}\right| \,. \tag{21}$$

Note that in the integrals defining the expressions (18) and (19) of $\hat{\psi}_u(K)$ and $\hat{\psi}_d(K)$, all the terms (Bogoliubov coefficients and coefficients of the $S$-matrix) involving a $d2$-subscript cancel when $\omega > \Omega$.

Defining $k_u = K - K_u$ and $k_d = K - K_d$ this yields for the upstream Fourier transform [instead of (18)],

$$\hat{\psi}_u(k_u < 0) = \widetilde{\mathcal{U}}_{u|\text{out}}(k_u)\Big[S_{u,u}\,\hat{b}_U + S_{u,d1}\,\hat{b}_{D1} + S_{u,d2}\,\hat{b}^\dagger_{D2}\Big]_{\omega = \omega_{u|\text{out}}(k_u)}$$
$$+ \quad \widetilde{\mathcal{W}}^*_{u|\text{in}}(k_u)\,\hat{b}^\dagger_U(\omega_{u|\text{in}}(-k_u)) \,, \tag{22}$$

$$\hat{\psi}_u(k_u > 0) = \widetilde{\mathcal{W}}^*_{u|\text{out}}(k_u)\Big[S^*_{u,u}\,\hat{b}^\dagger_U + S^*_{u,d1}\,\hat{b}^\dagger_{D1} + S^*_{u,d2}\,\hat{b}_{D2}\Big]_{\omega = \omega_{u|\text{out}}(-k_u)}$$
$$+ \quad \widetilde{\mathcal{U}}_{u|\text{in}}(k_u)\,\hat{b}_U(\omega_{u|\text{in}}(k_u)) \,, \tag{23}$$

and for the downstream Fourier transform [instead of (19)],

$$\hat{\psi}_d(k_d > 0) = \widetilde{\mathcal{U}}_{d1|\text{out}}(k_d)\Big[S_{d1,u}\,\hat{b}_U + S_{d1,d1}\,\hat{b}_{D1} + S_{d1,d2}\,\hat{b}^\dagger_{D2}\Big]_{\omega = \omega_{d1|\text{out}}(k_d)}$$

$$+ \quad \widetilde{\mathcal{U}}_{d2|\text{out}}(k_d) \Big[ S_{d2,u} \, \hat{b}_U + S_{d2,d1} \, \hat{b}_{D1} + S_{d2,d2} \, \hat{b}_{D2}^\dagger \Big]_{\omega = \omega_{d2|\text{out}}(k_d)}$$

$$+ \quad \widetilde{\mathcal{W}}_{d1|\text{in}}^*(k_d) \, \hat{b}_{D1}^\dagger(\omega_{d1|\text{in}}(-k_d))$$

$$+ \quad \widetilde{\mathcal{U}}_{d2|\text{in}}(k_d) \, \hat{b}_{D2}^\dagger(\omega_{d2|\text{in}}(k_d)) \,, \tag{24}$$

$$\hat{\psi}_d(k_d < 0) \;=\; \widetilde{\mathcal{W}}_{d1|\text{out}}^*(k_d) \Big[ S_{d1,u}^* \, \hat{b}_U^\dagger + S_{d1,d1}^* \, \hat{b}_{D1}^\dagger + S_{d1,d2}^* \, \hat{b}_{D2} \Big]_{\omega = \omega_{d1|\text{out}}(-k_d)}$$

$$+ \quad \widetilde{\mathcal{W}}_{d2|\text{out}}^*(k_d) \Big[ S_{d2,u}^* \, \hat{b}_U^\dagger + S_{d2,d1}^* \, \hat{b}_{D1}^\dagger + S_{d2,d2}^* \, \hat{b}_{D2} \Big]_{\omega = \omega_{d2|\text{out}}(-k_d)}$$

$$+ \quad \widetilde{\mathcal{U}}_{d1|\text{in}}(k_d) \, \hat{b}_{D1}(\omega_{d1|\text{in}}(k_d))$$

$$+ \quad \widetilde{\mathcal{W}}_{d2|\text{in}}^*(k_d) \, \hat{b}_{D2}(\omega_{d2|\text{in}}(-k_d)) \,. \tag{25}$$

All the terms in the expressions (22), (23), (24) and (25) depend on $k_u$ $(k_d)$ either directly, either *via* $\omega_\ell(\pm k_u)$ $(\omega_m(\pm k_d))$ where $\ell \in \{u|\text{in}, u|\text{out}\}$ $(m \in \{d1|\text{in}, d1|\text{out}, d2|\text{in}, d2|\text{out}\})$.

### 2.4.2   The case of the flat profile configuration

In this configuration – presented in Sec. 2.1.3 – the fact that $K_u = K_d \equiv K_0$ makes it possible to express (22), (23), (24) and (25) in terms of $k$ $(= k_u = k_d = K - K_0)$ only. It is then possible – and useful, see sec. 3 below – to regroup the momentum operators in Eqs. (22), (23), (24) and (25) in terms of $k < 0$ and $k > 0$ contributions, and to write

$$\hat{\psi}(k < 0) \;=\; \widetilde{\mathcal{U}}_{u|\text{out}}(k) \Big[ S_{u,u} \, \hat{b}_U + S_{u,d1} \, \hat{b}_{D1} + S_{u,d2} \, \hat{b}_{D2}^\dagger \Big]_{\omega = \omega_{u|\text{out}}(k)}$$

$$+ \quad \widetilde{\mathcal{W}}_{u|\text{in}}^*(k) \, \hat{b}_U^\dagger(\omega_{u|\text{in}}(-k))$$

$$+ \quad \widetilde{\mathcal{W}}_{d1|\text{out}}^*(k) \Big[ S_{d1,u}^* \, \hat{b}_U^\dagger + S_{d1,d1}^* \, \hat{b}_{D1}^\dagger + S_{d1,d2}^* \, \hat{b}_{D2} \Big]_{\omega = \omega_{d1|\text{out}}(-k)}$$

$$+ \quad \widetilde{\mathcal{W}}_{d2|\text{out}}^*(k) \Big[ S_{d2,u}^* \, \hat{b}_U^\dagger + S_{d2,d1}^* \, \hat{b}_{D1}^\dagger + S_{d2,d2}^* \, \hat{b}_{D2} \Big]_{\omega = \omega_{d2|\text{out}}(-k)}$$

$$+ \quad \widetilde{\mathcal{U}}_{d1|\text{in}}(k) \, \hat{b}_{D1}(\omega_{d1|\text{in}}(k))$$

$$+ \quad \widetilde{\mathcal{W}}_{d2|\text{in}}^*(k) \, \hat{b}_{D2}(\omega_{d2|\text{in}}(-k)) \,, \tag{26}$$

$$\hat{\psi}(k > 0) \;=\; \widetilde{\mathcal{W}}_{u|\text{out}}^*(k) \Big[ S_{u,u}^* \, \hat{b}_U^\dagger + S_{u,d1}^* \, \hat{b}_{D1}^\dagger + S_{u,d2}^* \, \hat{b}_{D2} \Big]_{\omega = \omega_{u|\text{out}}(-k)}$$

$$+ \quad \widetilde{\mathcal{U}}_{u|\text{in}}(k) \, \hat{b}_U(\omega_{u|\text{in}}(k))$$

$$+ \quad \widetilde{\mathcal{U}}_{d1|\text{out}}(k) \Big[ S_{d1,u} \, \hat{b}_U + S_{d1,d1} \, \hat{b}_{D1} + S_{d1,d2} \, \hat{b}_{D2}^\dagger \Big]_{\omega = \omega_{d1|\text{out}}(k)}$$

$$+ \quad \widetilde{\mathcal{U}}_{d2|\text{out}}(k) \Big[ S_{d2,u} \, \hat{b}_U + S_{d2,d1} \, \hat{b}_{D1} + S_{d2,d2} \, \hat{b}_{D2}^\dagger \Big]_{\omega = \omega_{d2|\text{out}}(k)}$$

$$+ \quad \widetilde{\mathcal{W}}_{d1|\text{in}}^*(k) \, \hat{b}_{D1}^\dagger(\omega_{d1|\text{in}}(-k))$$

$$+ \quad \widetilde{\mathcal{U}}_{d2|\text{in}}(k) \, \hat{b}_{D2}^\dagger(\omega_{d2|\text{in}}(k)) \,. \tag{27}$$

Each of the above expressions ($k < 0$ and $k > 0$) contains terms coming from both the subsonic and the supersonic regions. This procedure can induce a problem of normalization. As discussed in Sec. 4 after Eq. (92) this problem is easily solved by using an appropriate overall multiplicative factor which we do not include for readability. Furthermore, the problem disappears when one considers normalized quantity such as $g_2(K, Q)$ defined below [Eq. (29)].

# 3 Momentum correlations in the presence of a sonic horizon

The momentum-momentum correlation signal is embodied in the function

$$G_2(K, Q) = \langle :\hat{N}(K)\hat{N}(Q): \rangle - \langle \hat{N}(K) \rangle \langle \hat{N}(Q) \rangle \, , \tag{28}$$

where $\hat{N}(K) = \hat{\psi}^\dagger(K)\hat{\psi}(K)$. We also consider in some details the normalized quantity

$$g_2(K, Q) = \frac{\langle :\hat{N}(K)\hat{N}(Q): \rangle}{\langle \hat{N}(K) \rangle \langle \hat{N}(Q) \rangle} \, . \tag{29}$$

In the definitions (28) and (29) the normally ordered product eliminates the diagonal shot noise contribution.

The computation of the two-body momentum correlation (28) in the generic case is quite tedious because the upstream and downstream Fourier transform are different, as explained in Sec. 2.4.1. Besides the formulas assume different forms depending of the values of $K$ and $Q$ relative to $K_u$ and $K_d$, cf. Eqs. (22), (23), (24) and (25). As a result, one has to consider nine different cases. Although we will treat all the black-hole configurations presented in Sec. 2.1 (plots encompassing all the different cases for the waterfall, $\delta$ peak and flat profile configurations are given in Figs. 4 and 5), to simplify the presentation we give here the explicit results in the "flat profile" configuration where the background density is a uniform plane wave (cf. Sec. 2.1.3). In this case one has to consider only 4 cases: the four quadrants in the $(k, q)$ plane (where $k = K - K_0$ and $q = Q - K_0$) and one can rewrite $G_2$ in terms of $k$ and $q$

$$G_2(k, q) = \langle \hat{\psi}^\dagger(k)\hat{\psi}^\dagger(q)\hat{\psi}(k)\hat{\psi}(q) \rangle - \langle \hat{\psi}^\dagger(k)\hat{\psi}(k) \rangle \langle \hat{\psi}^\dagger(q)\hat{\psi}(q) \rangle \, , \tag{30}$$

and then perform explicit computations using the expressions (26) and (27).

The theoretical evaluation of the momentum correlation function is performed in order to match with an experimental detection scheme which consists of opening the trap and letting the elementary excitations be converted into particles expelled from both ends of the condensate, according to a process known as "phonon evaporation" [35]. If this process is assumed to be gentle and adiabatic, each elementary excitation is converted into a single particle. More precisely, one has, in this case, for the positive norm $u$ and $d1$ modes:

$$\mathcal{U}_\ell(\omega) \to |\partial\omega/\partial q_\ell|^{-1/2} \quad \text{and} \quad \mathcal{W}_\ell(\omega) \to 0 \, , \quad \text{for} \quad \ell \in \{u|\text{in}, u|\text{out}, d1|\text{in}, d1|\text{out}\} \, ; \tag{31}$$

while for the negative norm $d2$ modes:

$$\mathcal{U}_\ell(\omega) \to 0 \quad \text{and} \quad \mathcal{W}_\ell(\omega) \to |\partial\omega/\partial q_\ell|^{-1/2} \, , \quad \text{for} \quad \ell \in \{d2|\text{in}, d2|\text{out}\} \, . \tag{32}$$

With the normalization (12), this corresponds, for instance for a positive norm mode, to a perfect transmutation of an elementary excitation into a particle state which carries a unit current. For the "tilde Bogoliubov coefficients" (20), the prescription (31) yields

$$\widetilde{\mathcal{U}}_\ell(q) \to |\partial\omega_\ell/\partial q|^{1/2} \quad \text{and} \quad \widetilde{\mathcal{W}}_\ell(q) \to 0 \, , \tag{33}$$

for the positive norm states ($\ell \in \{u|\text{in}, u|\text{out}, d1|\text{in}, d1|\text{out}\}$) whereas one gets

$$\widetilde{\mathcal{U}}_\ell(q) \to 0 \quad \text{and} \quad \widetilde{\mathcal{W}}_\ell(q) \to |\partial\omega_\ell/\partial q|^{1/2} \, , \tag{34}$$

for the negative norm modes ($\ell \in \{d2|\text{in}, d2|\text{out}\}$).

As demonstrated in Ref. [22], after an adiabatic opening of the trapping potential, a measure of the velocity distribution of the emitted particles gives access to the momentum distribution within the condensate and to the correlators defined in Eqs. (28) and (29). The process can be sudden, in which case the adiabatic hypothesis breaks down. More precisely, if $t_{\text{open}}$ is the characteristic time during which the trap is opened and an elementary excitation is converted into a real particle, the adiabatic approximation fails for excitations of energy $\omega$ verifying: $\omega^{-1} \gg t_{\text{open}}$: for low lying excitations (those for which $\omega \to 0$) the opening of the trap always seems abrupt [35]. A quantitative study of this phenomenon will be presented in a future publication [36].

An outline of the complications introduced by non adiabatic effects is postponed to Section (3.3), but until then we give the results after an adiabatic expansion, in which case, owing to (33) and (34), the expression of $\hat{\psi}$ given in (26), (27) reduces to

$$
\begin{aligned}
\hat{\psi}(k < 0) \;=\; & \widetilde{\mathcal{U}}_{u|\text{out}}\Big[S_{u,u}\,\hat{b}_U + S_{u,d1}\,\hat{b}_{D1} + S_{u,d2}\,\hat{b}_{D2}^\dagger\Big]_{\omega = \omega_{u|\text{out}}(k)} \\
+ \;& \widetilde{\mathcal{W}}_{d2|\text{out}}^*\Big[S_{d2,u}^*\,\hat{b}_U^\dagger + S_{d2,d1}^*\,\hat{b}_{D1}^\dagger + S_{d2,d2}^*\,\hat{b}_{D2}\Big]_{\omega = \omega_{d2|\text{out}}(-k)} \\
+ \;& \widetilde{\mathcal{U}}_{d1|\text{in}}\hat{b}_{D1}(\omega_{d1|\text{in}}(k)) + \widetilde{\mathcal{W}}_{d2|\text{in}}^*\hat{b}_{D2}(\omega_{d2|\text{in}}(-k)) \, ,
\end{aligned}
\tag{35}
$$

and

$$
\begin{aligned}
\hat{\psi}(k > 0) \;=\; & \widetilde{\mathcal{U}}_{d1|\text{out}}\Big[S_{d1,u}\,\hat{b}_U + S_{d1,d1}\,\hat{b}_{D1} + S_{d1,d2}\,\hat{b}_{D2}^\dagger\Big]_{\omega = \omega_{d1|\text{out}}(k)} \\
+ \;& \widetilde{\mathcal{U}}_{u|\text{in}}\hat{b}_U(\omega_{u|\text{in}}(k)) \, .
\end{aligned}
\tag{36}
$$

In expressions (35) and (36) the "tilde Bogoliubov coefficients" which are non zero assume the limiting values (33) and (34).

## 3.1 Zero temperature

Let us first consider the case where the system is initially in its vacuum state, i.e., the vacuum of excitations. This is the zero temperature case. We find for the one-body momentum distribution:

$$
\begin{aligned}
\langle \hat{N}(k < 0) \rangle =\; & \Big(|\widetilde{\mathcal{U}}_{u|\text{out}}|^2 |S_{u,d2}|^2\Big)_{\omega_{u|\text{out}}(k)} \times \delta\left(\omega_{u|\text{out}}(k) - \omega_{u|\text{out}}(k)\right) \\
& + |\widetilde{\mathcal{W}}_{d2|\text{out}}|^2\left(|S_{d2,u}|^2 + |S_{d2,d1}|^2\right)_{\omega_{d2|\text{out}}(-k)} \times \delta\left(\omega_{d2|\text{out}}(-k) - \omega_{d2|\text{out}}(-k)\right)
\end{aligned}
\tag{37}
$$

and

$$\langle \hat{N}(k > 0) \rangle = \Big(|\widetilde{\mathcal{U}}_{d1|\text{out}}|^2 |S_{d1,d2}|^2\Big)_{\omega_{d1|\text{out}}(k)} \times \delta\left(\omega_{d1|\text{out}}(k) - \omega_{d1|\text{out}}(k)\right) \, . \tag{38}$$

These relations can be cast under the form

$$\langle \hat{N}(k) \rangle = \mathcal{N}(k) \times \delta(k - k) \, , \tag{39}$$

where

$$\mathcal{N}(k) = \begin{cases} |S_{u,d2}|^2_{\omega_{u|\text{out}}(k)} + \left(|S_{d2,u}|^2 + |S_{d2,d1}|^2\right)_{\omega_{d2|\text{out}}(-k)} & \text{for } k < 0 , \\ |S_{d1,d2}|^2_{\omega_{d1|\text{out}}(k)} & \text{for } k > 0 . \end{cases} \tag{40}$$

Note that within the adiabatic approximation, the $T = 0$ momentum signal (40) would cancel in the absence of horizon, since the $d2$ mode and all the corresponding elements of the $S$-matrix would disappear in this case. This is confirmed by the study of Sec. 4.

In expression (39), the $\delta$-peak contribution is singular: one has a $\delta(0)$ [as in (37) and (38)]. This is due to the schematic nature of the rules R1 and R2 defined in section 2.4.1 for the Fourier transform. The more rigorous local Fourier transform in terms of a window function –explained in the first part of Sec. 2.4.1– yields a finite contribution, as shown in Appendix A: see, e.g., Eqs. (108) and (109) which are the rigorous versions of Eqs. (37) and (38).

Our main interest in this work is the study of the correlation signal in momentum space, that is of the two-points correlation functions $G_2$ and $g_2$ defined in Eqs. (28) and (29). The most robust signals are those present even at $T = 0$. These are

$$G_2(k < 0, q < 0) = \left[|S_{u,d2}|^2_{\omega_{u|\text{out}}(k)} + \left(|S_{d2,u}|^2 + |S_{d2,d1}|^2\right)\big|_{\omega_{d2|\text{out}}(-k)}\right]^2 \delta^2(k - q)$$
$$+ \left[|\widetilde{\mathcal{U}}_{u|\text{out}}|^2 |\widetilde{\mathcal{W}}_{d2|\text{out}}|^2 \left|S^*_{u,d2} S_{d2,d2}\right|^2 \delta^2(\omega_{u|\text{out}}(k) - \omega_{d2|\text{out}}(-q)) + (k \leftrightarrow q)\right], \tag{41}$$

$$G_2(k > 0, q > 0) = |S_{d1,d2}|^2_{\omega_{d1|\text{out}}(k)} \delta^2(k - q) , \tag{42}$$

$$G_2(k < 0, q > 0) = |\widetilde{\mathcal{U}}_{u|\text{out}}|^2 |\widetilde{\mathcal{U}}_{d1|\text{out}}|^2 \left|S^*_{u,d2} S_{d1,d2}\right|^2 \delta^2(\omega_{u|\text{out}}(k) - \omega_{d1|\text{out}}(q))$$
$$+ |\widetilde{\mathcal{W}}_{d2|\text{out}}|^2 |\widetilde{\mathcal{U}}_{d1|\text{out}}|^2 \left|S^*_{d2,d2} S_{d1,d2}\right|^2 \delta^2(\omega_{d2|\text{out}}(-k) - \omega_{d1|\text{out}}(q)). \tag{43}$$

As already noted for the one-body signal, a first obvious outcome of this computation is that the $T = 0$ correlations in momentum space disappear in the absence of sonic horizon, since in this case the $d2$ mode does not exist and all the corresponding elements of the $S$ matrix cancel in (41), (42) and (43).

Looking more in detail into the results, one sees that in the $(k < 0, q < 0)$ and in the $(k > 0, q > 0)$ sectors one has terms with a $\delta^2(k - q)$ correlation which we henceforth denote as diagonal. These terms are simply of the form $\mathcal{N}^2(k)\delta^2(k - q)$. Again, the occurrence of the highly singular squared $\delta$ distribution in the expressions (41), (42) and (43) is an artifact of our schematic Fourier transform rules R1 and R2 which disappears when one considers windowed Fourier transforms, see Appendix A.

Besides the diagonal term, one has correlation lines along the $u|\text{out} - d2|\text{out}$ (Hawking quanta-partner), $d2|\text{out} - d1|\text{out}$ and $u|\text{out} - d1|\text{out}$ channels. All these correlations are also present *mutatis mutandis* in the density fluctuation sector [14, 15, 18, 25]. As in the interpretation of black-hole radiation first given by Hawking [3], the existence of these correlations is an indication of the fact that the vacuum of the ingoing modes is not the same as the vacuum of outgoing ones, and this results in spontaneous emission of pairs of outgoing quasi-particles even in the absence of incoming ones [37, 38]. This is possible even in our stationary setting because the number of created quasiparticles of positive energy equals the number negative energy ones, i.e. energy is conserved [25, 39].

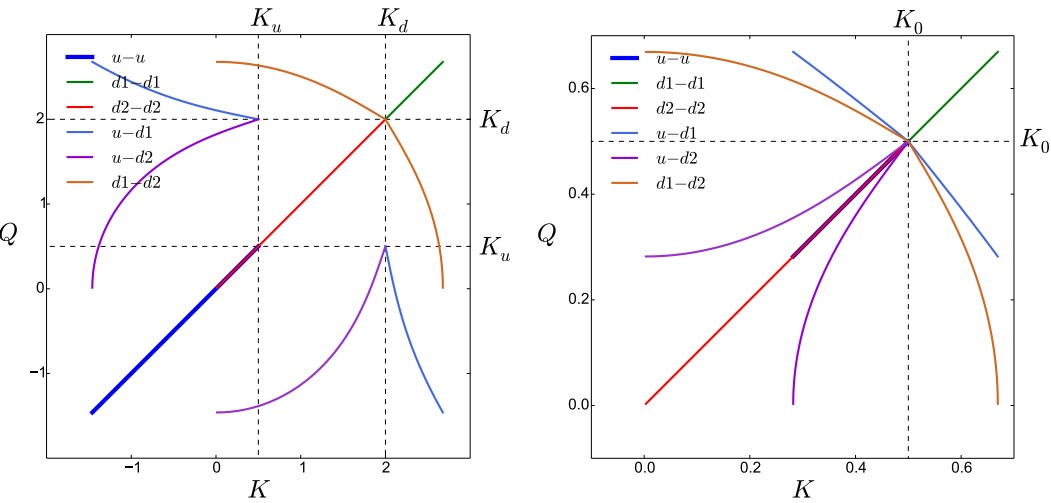

Figure 4: Zero temperature momentum space correlation lines from Eqs. (41), (42) and (43). These curves correspond to the loci of points with finite value of the two body momentum correlation function $G_2(K, Q)$. They are labeled with the names of the modes of correlated momenta, for instance the "$u - d1$" curve corresponds to the line of correlation between the $u|$out and the $d1|$out modes. The left plot displays the results for a waterfall configuration with $M_u = 0.5$ and $M_d = 4$. The right plot displays the results for a flat profile configuration with the same values of $M_u$ and $M_d$. The momenta are expressed in units of $\xi_u^{-1}$.

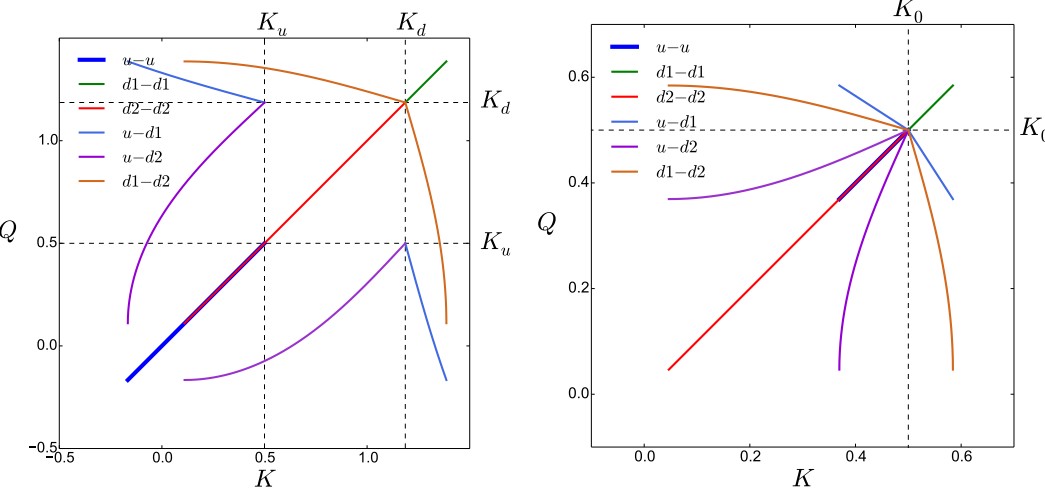

Figure 5: Same as Fig. 4 for a $\delta$ peak configuration (left plot) and flat profile (right plot) having both the same asymptotic Mach numbers $M_u = 0.5$ and $M_d = 1.827$.

The results corresponding to the zero temperature correlation signals (41), (42) and (43) are displayed in Figs. 4 and 5. In figure 4, the left plot presents the results for a waterfall configuration with $M_u = 0.5$; this imposes $M_d = 4$. The results for the flat profile configuration with the same values of $M_u$ and $M_d$ are displayed in the right plot. As noted at the end of Sec. 2.1.3, it is not possible to realize a waterfall and a $\delta$ peak profile having the same values of $M_u$ and $M_d$. Hence we compare in figure 5 the results for a $\delta$ peak configuration (left plot) with $M_u = 0.5$ (this imposes $M_d = 1.827$) with the ones for a flat profile configuration having the same asymptotic Mach numbers (right plot).

The graphical rule for drawing the correlation lines in these figures is simple. Let's consider the $u|\text{out} - d2|\text{out}$ correlation line for instance. From Eq. (41) one sees that the relative wave vectors $k$ and $q$ are correlated when $\omega_{u|\text{out}}(k) = \omega_{d2|\text{out}}(-q)$, and also along the curve obtained by exchanging the roles of $k$ and $q$. This corresponds to the two curves $q = -q_{d2|\text{out}}(\omega_{u|\text{out}}(k))$ and $q = q_{u|\text{out}}(\omega_{d2|\text{out}}(-k))$. Going to the absolute wave vectors $K$ and $Q$, these correlation lines correspond to the curves $Q - K_d = -q_{d2|\text{out}}(\omega_{u|\text{out}}(K - K_u))$ and $Q - K_u = q_{u|\text{out}}(\omega_{d2|\text{out}}(-K + K_d))$. These two curves are symmetrical with respect to the diagonal. In the case of the flat profile configuration they meet on the diagonal at the point of coordinate $(K_u, K_d = K_u)$.

Figures 4 and 5 indicate the location of the relevant correlation signal, but not its amplitude. The theoretical evaluation of this amplitude can be done either using the schematic rules R1 and R2 [but then yields to singular expressions as in (41), (42) and (43)] either using windowed Fourier transforms (but then depends on the specific choice of the windows, cf. Appendix A). One can circumvent these difficulties and obtain a non-ambiguous result by working with the rescaled $g_2$ function (29), as explained now.

Let us first consider "diagonal correlation terms" which are intra-channel correlation in the $u|\text{out}$, $d1|\text{out}$ and $d2|\text{out}$ channels corresponding to the diagonal lines in Figs. 4 and 5. These diagonal terms are conveniently isolated by studying correlation functions of the type

$$g_2(K, K)_{u|\text{out}} = \frac{G_2(K, K)_{u|\text{out}}}{\langle \hat{N}(K) \rangle^2_{u|\text{out}}} + 1 \, , \tag{44}$$

where $\langle \hat{N}(K) \rangle_{u|\text{out}}$ and $G_2(K, K)_{u|\text{out}}$ are the $u|\text{out}$ contributions to $\langle \hat{N}(K) \rangle$ and to the diagonal part of $G_2(K, K)$. One obtains straightforwardly

$$g_2(K, K)_{u|\text{out}} = 2 \, , \tag{45}$$

a result which follows from Wick's theorem and which is also valid for the other channels, even at finite temperature, for non-adiabatic opening of the trap, and also when considering Fourier transforms less schematic than in Eqs. (18) and (19) (cf. Appendix A).

At zero temperature, in the adiabatic regime we consider in the present subsection, the only non-diagonal contributions to $G_2$ are of the type $u|\text{out}–d2|\text{out}$, $u|\text{out}–d1|\text{out}$ and $d1|\text{out}–d2|\text{out}$. One considers here for instance correlation functions of the type (with obvious notations)

$$g_2(K, Q)_{u|\text{out}-d2|\text{out}} = \frac{G_2(K, Q)_{u|\text{out}-d2|\text{out}}}{\langle \hat{N}(K) \rangle_{u|\text{out}} \langle \hat{N}(Q) \rangle_{d2|\text{out}}} + 1 \, . \tag{46}$$

From expressions (37) and (41) one gets

$$g_2(K, Q)_{u|\text{out}-d2|\text{out}} = \frac{|S_{d2,d2}|^2}{|S_{d2,u}|^2 + |S_{d2,d1}|^2} + 1 = \frac{2|S_{d2,d2}|^2 - 1}{|S_{d2,d2}|^2 - 1} \, . \tag{47}$$

For obtaining the r.h.s. of Eq. (47) we used the pseudo-unitarity condition (14) and also the fact that

$$\frac{\delta^2(\omega_{u|\text{out}}(k) - \omega_{d2|\text{out}}(-q))}{\delta\left(\omega_{u|\text{out}}(k) - \omega_{u|\text{out}}(k)\right)\delta\left(\omega_{d2|\text{out}}(-k) - \omega_{d2|\text{out}}(-k)\right)} = 1 \ . \tag{48}$$

While this relation is easily verified in the schematic framework used in the main text (which originates from the rules R1 and R2 of Sec. 2.4.1), it appears less straightforward when studying the more rigorously defined local Fourier transform (see Appendix A). In this case, the equivalent of Eq. (48) is Eq. (124) and is equal to unity only if Eqs. (126) are verified, i.e., if the window functions fulfill specific conditions, the physical content of which is discussed in Appendix A. We will assume that these conditions are met in the following (or equivalently we keep on using the schematic rules R1 and R2 of Sec. 2.4.1). For the other inter-channel correlators the normalized two-body functions read:

$$g_2(K, Q)_{u|\text{out}-d1|\text{out}} = 2 \ , \tag{49}$$

and

$$g_2(K, Q)_{d1|\text{out}-d2|\text{out}} = \frac{2|S_{d2,d2}|^2 - 1}{|S_{d2,d2}|^2 - 1} \ . \tag{50}$$

The study of $g_2$ is of interest because the occurrence of entanglement and the quantum nature of the Hawking process can be tested through the violation of the Cauchy-Schwarz inequality, as recently studied in a similar context in Refs. [32, 33, 40–46]. For instance, the Cauchy-Schwarz inequality is violated along the characteristic Hawking quanta–partner correlation lines $u|$out–$d2|$out of Figs. 4 or 5 if (see, e.g., [47])

$$g_2(K, Q)\Big|_{u|\text{out}-d2|\text{out}} > \sqrt{g_2(K, K)\Big|_{u|\text{out}} \times g_2(Q, Q)\Big|_{d2|\text{out}}} = 2 \ . \tag{51}$$

As already noticed after Eq. (45), the right-hand side of inequality (51) is equal to 2 for all temperature. From expression (47) one sees that the Cauchy-Schwarz inequality is violated at $T = 0$ along the $u|$out–$d2|$out correlation line for those values of $K$ and $Q$ such that, at energy $\omega = \omega_{u|\text{out}}(k) = \omega_{d2|\text{out}}(-q)$, one has $|S_{d2,d2}(\omega)| > 1$. $S_{d2,d2}$ is the scattering amplitude from the $d2|$in mode towards the $d2|$out mode; its modulus can be larger than unity without violating the pseudo-unitarity condition (14). $|S_{d2,d2}(\omega)|$ diverges as $\omega^{-1/2}$ when $\omega \to 0$ [25] and the Cauchy-Schwarz inequality is thus always violated for $\omega > 0$. The same holds true for the violation of the Cauchy-Schwarz inequality along the $d1|$out–$d2|$out channels at $T = 0$ since the formula (50) yields for $g_2(K, Q)_{d1|\text{out}-d2|\text{out}}$ the same result than (47) does for $g_2(K, Q)_{u|\text{out}-d2|\text{out}}$. From Eq. (49) it is clear that the Cauchy-Schwarz inequality is not violated at $T = 0$ along the $u|$out–$d1|$out correlation channel.

A remark should be made here concerning the experimental detection process. Our theoretical analysis corresponds to windowed upstream and downstream momentum detection. It is thus theoretically possible to distinguish an upstream and a downstream component in the signal in momentum space. This is the reason why it is legitimate to identify, for instance a $u|$out $- d2|$out component in the total $G_2(K, Q)$, or a $u|$out component in the total $\langle \hat{N}(K) \rangle$, as done in Eq. (46). However, some apparatuses may mix the upstream and downstream signals in their detection scheme. In this case, the overlap in momentum space of the domain of existence of the $u|$out and $d2|$out signals forbids the use of a definition as precise as (46). In this case, a clear separation can only be done for the momenta located outside of the overlap

region. In the waterfall configuration represented in the left plot of Fig. 4 for instance, with such a detection apparatus, one cannot study the $u|\text{out} - d2|\text{out}$ correlation signal in regions where the $d2|\text{out}$ and $u|\text{out}$ momenta overlap, i.e, when $K$ or $Q$ lie in a segment for which the blue and red diagonal curves of the figure overlap. In this respect, such an apparatus would not have access to the $u|\text{out} - d2|\text{out}$ correlation signal at all for the flat profile configuration, since in this case the region of momenta issued from the $u|\text{out}$ channel is completely contained in the region of momenta issued from the $d2|\text{out}$ channel. To recall this possible experimental issue, when we plot below quantities such as $g_2(K,Q)\big|_{u|\text{out}-d2|\text{out}}$ or $g_2(K,Q)\big|_{u|\text{out}-d1|\text{out}}$, we shade the region of momenta where an ambiguity is possible (cf. Figs. 6 and 7). We stress however that not all experimental detection schemes have to be subject to this drawback.

## 3.2 Finite temperature

The analog stationary black hole configuration we consider is thermodynamically unstable, and cannot support a thermal state. However, a thermal-like occupation of the states can be defined by considering a time dependent process of formation of the horizon starting from a uniform thermal subsonic uniform configuration, following the procedure already considered in Ref. [18] (see also [17]): The condensate has initially a uniform density $n_u$, a flow velocity $V_u$ and a sound velocity $c_u$ and is at thermal equilibrium at temperature $T$ in the moving frame. The horizon is then adiabatically switched on, either by ramping down the scattering length in the downstream region – leading to a flat profile configuration –, or by ramping up an external potential – leading to a waterfall or a delta peak configuration.

Then, one obtains occupation numbers $n_U(\omega)$, $n_{D1}(\omega)$ and $n_{D2}(\omega)$ for each of the scattering modes. For instance $n_U(\omega) = n_{\text{th}}[\omega_{\text{B}}(q_{u|\text{in}}(\omega))]$, where $n_{\text{th}}(\Omega) = (\exp(\Omega/T) - 1)^{-1}$ is the thermal Bose occupation factor, and $\omega_{\text{B}}(q)$ is the Bogoliubov dispersion relation (6) in the moving frame (with here $c \to c_u$ and $\xi \to \xi_u$ in expression (6)). This procedure leads for the other occupation numbers $n_{D1}(\omega) = n_{\text{th}}[\omega_{\text{B}}(q_{d1|\text{in}}(\omega))]$ and $n_{D2}(\omega) = n_{\text{th}}[\omega_{\text{B}}(q_{d2|\text{in}}(\omega))]$, it corresponds to the experimental situation where an external potential is swept at constant velocity through a condensate initially at rest [24, 48].

Note that one could choose other prescriptions for defining the occupation of the scattering modes. For instance the initial state (uniform density $n_u$ with uniform velocity $V_u$) could be in thermal equilibrium in the frame of the obstacle (and not in the frame where the condensate is at rest, as considered above); this would modify the explicit expression of the $n_L$'s in (52). Precisely one would have in this case: $n_U(\omega) = n_{\text{th}}(\omega)$, $n_{D1}(\omega) = n_{\text{th}}(\omega_{u|\text{out}}(q_{d1|\text{in}}(\omega)))$ and $n_{D1}(\omega) = n_{\text{th}}(\omega_{u|\text{out}}(-q_{d2|\text{in}}(\omega)))$. In the following we just use the contraction rules

$$\begin{aligned}
\langle \hat{b}_L(\omega)\hat{b}^\dagger_{L'}(\omega')\rangle &= [1 + n_L(\omega)]\,\delta_{L,L'}\,\delta(\omega - \omega')\,, \\
\langle \hat{b}^\dagger_L(\omega)\hat{b}_{L'}(\omega')\rangle &= n_L(\omega)\,\delta_{L,L'}\,\delta(\omega - \omega')\,,
\end{aligned} \tag{52}$$

without specifying the expressions of the $n_L$'s, and the formulas written below are thus generally valid.

In the adiabatic regime where Eqs. (33) and (34) hold, we find, for negative $k$:

$$\langle \hat{N}(k<0) \rangle = |\widetilde{\mathcal{U}}_{u|\text{out}}|^2 \left( |S_{u,u}|^2 n_U + |S_{u,d1}|^2 n_{D1} + |S_{u,d2}|^2 (1+n_{D2}) \right)_{\omega_{u|\text{out}}(k)}$$

$$\times\, \delta \left( \omega_{u|\text{out}}(k) - \omega_{u|\text{out}}(k) \right)$$

$$+\, |\widetilde{\mathcal{W}}_{d2|\text{out}}|^2 \left( |S_{d2,u}|^2 (1+n_U) + |S_{d2,d1}|^2 (1+n_{D1}) + |S_{u,d2}|^2 n_{D2} \right)_{\omega_{d2|\text{out}}(-k)}$$

$$\times\, \delta \left( \omega_{d2|\text{out}}(-k) - \omega_{d2|\text{out}}(-k) \right)$$

$$+\, |\widetilde{\mathcal{U}}_{d1|\text{in}}|^2 n_{D1}(\omega_{d1|\text{in}}(k)) \times \delta \left( \omega_{d1|\text{in}}(k) - \omega_{d1|\text{in}}(k) \right)$$

$$+\, |\widetilde{\mathcal{W}}_{d2|\text{in}}|^2 n_{D2}(\omega_{d2|\text{in}}(-k)) \times \delta \left( \omega_{d2|\text{in}}(-k) - \omega_{d2|\text{in}}(-k) \right) \;, \tag{53}$$

and, for positive $k$:

$$\langle \hat{N}(k>0) \rangle = |\widetilde{\mathcal{U}}_{d1|\text{out}}|^2 \left( |S_{d1,u}|^2 n_U + |S_{d1,d1}|^2 n_{D1} + |S_{d1,d2}|^2 (1+n_{D2}) \right)_{\omega_{d1|\text{out}}(k)}$$

$$\times\, \delta \left( \omega_{d1|\text{out}}(k) - \omega_{d1|\text{out}}(k) \right) \tag{54}$$

$$+\, |\widetilde{\mathcal{U}}_{u|\text{in}}|^2 n_U(\omega_{u|\text{in}}(k)) \times \delta \left( \omega_{u|\text{in}}(k) - \omega_{u|\text{in}}(k) \right).$$

As in the zero temperature case, formulas (53) and (54) can be cast under the form (39), with here

$$\mathcal{N}(k<0) = \left( |S_{u,u}|^2 n_U + |S_{u,d1}|^2 n_{D1} + |S_{u,d2}|^2 (1+n_{D2}) \right)_{\omega_{u|\text{out}}(k)}$$

$$+\, \left( |S_{d2,u}|^2 (1+n_U) + |S_{d2,d1}|^2 (1+n_{D1}) + |S_{d2,d2}|^2 n_{D2} \right)_{\omega_{d2|\text{out}}(-k)} \tag{55}$$

$$+\, n_{D1}(\omega_{d1|\text{in}}(k)) + n_{D2}(\omega_{d2|\text{in}}(-k)) \;.$$

and

$$\mathcal{N}(k>0) = \left( |S_{d1,u}|^2 n_U + |S_{d1,d1}|^2 n_{D1} + |S_{d1,d2}|^2 (1+n_{D2}) \right)_{\omega_{d1|\text{out}}(k)}$$

$$+\, n_U(\omega_{u|\text{in}}(k)) \;. \tag{56}$$

Then, one obtains for the momentum correlation:

$$G_2(k<0, q<0) = \mathcal{N}^2(k<0)\, \delta^2(k-q)$$

$$+\; \left[ |\widetilde{\mathcal{U}}_{u|\text{out}}|^2 |\widetilde{\mathcal{W}}_{d2|\text{out}}|^2 \left| S_{u,u}^* S_{d2,u} n_U + S_{u,d1}^* S_{d2,d1} n_{D1} + S_{u,d2}^* S_{d2,d2} (1+n_{D2}) \right|^2 \times \right.$$

$$\delta^2(\omega_{u|\text{out}}(k) - \omega_{d2|\text{out}}(-q))$$

$$+\; |\widetilde{\mathcal{U}}_{u|\text{out}}|^2 |\widetilde{\mathcal{U}}_{d1|\text{in}}|^2 |S_{u,d1}|^2 n_{D1}^2 \delta^2(\omega_{u|\text{out}}(k) - \omega_{d1|\text{in}}(q))$$

$$+\; |\widetilde{\mathcal{U}}_{u|\text{out}}|^2 |\widetilde{\mathcal{W}}_{d2|\text{in}}|^2 |S_{u,d2}|^2 n_{D2}(1+n_{D2}) \delta^2(\omega_{u|\text{out}}(k) - \omega_{d2|\text{in}}(-q))$$

$$+\; |\widetilde{\mathcal{W}}_{d2|\text{out}}|^2 |\widetilde{\mathcal{U}}_{d1|\text{in}}|^2 |S_{d2,d1}|^2 n_{D1}(1+n_{D1}) \delta^2(\omega_{d2|\text{out}}(-k) - \omega_{d1|\text{in}}(q))$$

$$+\; |\widetilde{\mathcal{W}}_{d2|\text{out}}|^2 |\widetilde{\mathcal{W}}_{d2|\text{in}}|^2 |S_{d2,d2}|^2 n_{D2}^2 \delta^2(\omega_{d2|\text{out}}(-k) - \omega_{d2|\text{in}}(-q))$$

$$+\; \left. (k \leftrightarrow q) \right] \;, \tag{57}$$

$$G_2(k>0, q>0) = \mathcal{N}^2(k>0)\, \delta^2(k-q)$$

$$+ \quad \left[ |\widetilde{\mathcal{U}}_{u|\text{in}}|^2 |\widetilde{\mathcal{U}}_{d1|\text{out}}|^2 |S_{d1,u}|^2 n_U^2 \; \delta^2(\omega_{d1|\text{out}}(k) - \omega_{u|\text{in}}(q)) + (k \leftrightarrow q) \right] \; . \tag{58}$$

In the first term of the last line of this formula there is no ambiguity: $|S_{d1,u}|^2$ and $n_U$ have to be evaluated at $\omega_{d1|\text{out}}(k) = \omega_{u|\text{in}}(q)$, $|\widetilde{\mathcal{U}}_{u|\text{in}}|^2$ has to be evaluated at $q$ and $|\widetilde{\mathcal{U}}_{d1|\text{out}}|^2$ has to be evaluated at $k$.

$$G_2(k < 0, q > 0) =$$

$$|\widetilde{\mathcal{U}}_{u|\text{out}}|^2 |\widetilde{\mathcal{U}}_{d1|\text{out}}|^2 \left| S_{u,u}^* S_{d1,u} n_U + S_{u,d1}^* S_{d1d1} n_{D1} + S_{u,d2}^* S_{d1,d2}(1 + n_{D2}) \right|^2$$
$$\delta^2(\omega_{u|\text{out}}(k) - \omega_{d1|\text{out}}(q))$$

$$+ \quad |\widetilde{\mathcal{W}}_{d2|\text{out}}|^2 |\widetilde{\mathcal{U}}_{d1|\text{out}}|^2 \left| S_{d2,u}^* S_{d1,u} n_U + S_{d2,d1}^* S_{d1,d1} n_{D1} + S_{d2,d2}^* S_{d1,d2}(1 + n_{D2}) \right|^2$$
$$\delta^2(\omega_{d2|\text{out}}(-k) - \omega_{d1|\text{out}}(q))$$

$$+ \quad |\widetilde{\mathcal{U}}_{u|\text{out}}|^2 |\widetilde{\mathcal{U}}_{u|\text{in}}|^2 |S_{u,u}|^2 n_U^2 \delta^2(\omega_{u|\text{out}}(k) - \omega_{u|\text{in}}(q))$$

$$+ \quad |\widetilde{\mathcal{U}}_{d1|\text{out}}|^2 |\widetilde{\mathcal{U}}_{d1|\text{in}}|^2 |S_{d1,d1}|^2 n_{D1}^2 \delta^2(\omega_{d1|\text{in}}(k) - \omega_{d1|\text{out}}(q))$$

$$+ \quad |\widetilde{\mathcal{W}}_{d2|\text{out}}|^2 |\widetilde{\mathcal{U}}_{u|\text{in}}|^2 |S_{d2,u}|^2 n_U(1 + n_U) \delta^2(\omega_{d2|\text{out}}(-k) - \omega_{u|\text{in}}(q))$$

$$+ \quad |\widetilde{\mathcal{U}}_{d1|\text{out}}|^2 |\widetilde{\mathcal{W}}_{d2|\text{in}}|^2 |S_{d1,d2}|^2 n_{D2}(1 + n_{D2}) \delta^2(\omega_{d2|\text{in}}(-k) - \omega_{d1|\text{out}}(q)) \; . \tag{59}$$

We remark that at $T \neq 0$ there appear in-out correlators, which were absent at $T = 0$. We shall focus here on the most robust correlators, those already present at $T = 0$, and evaluate the intensity of the correlation signal along these lines by using the normalized correlation function $g_2$ which is window independent. The Hawking quanta-partner correlator (47) gets modified to

$$g_2(K,Q)_{u|\text{out}-d2|\text{out}} = \frac{\left| S_{u,u}^* S_{d2,u} n_U + S_{u,d1}^* S_{d2,d1} n_{D1} + S_{u,d2}^* S_{d2,d2}(1 + n_{D2}) \right|^2}{\mathcal{N}_{u|\text{out}}(k) \, \mathcal{N}_{d2|\text{out}}(q)} + 1 \; , \tag{60}$$

where [from (55)]

$$\mathcal{N}_{u|\text{out}}(k) = \left( |S_{u,u}|^2 n_U + |S_{u,d1}|^2 n_{D1} + |S_{u,d2}|^2(1 + n_{D2}) \right)_{\omega_{u|\text{out}}(k)} \; , \tag{61}$$

and

$$\mathcal{N}_{d2|\text{out}}(k) = \left( |S_{d2,u}|^2(1 + n_U) + |S_{d2,d1}|^2(1 + n_{D1}) + |S_{d2,d2}|^2 n_{D2} \right)_{\omega_{d2|\text{out}}(-k)}$$
$$= \left( -1 + |S_{d2,u}|^2 n_U + |S_{d2,d1}|^2 n_{D1} + |S_{d2,d2}|^2(1 + n_{D2}) \right)_{\omega_{d2|\text{out}}(-k)} \; . \tag{62}$$

The $u|\text{out} - d1|\text{out}$ correlator (49) becomes

$$g_2(K,Q)_{u|\text{out}-d1|\text{out}} = \frac{\left| S_{u,u}^* S_{d1,u} n_U + S_{u,d1}^* S_{d1,d1} n_{D1} + S_{u,d2}^* S_{d1,d2}(1 + n_{D2}) \right|^2}{\mathcal{N}_{u|\text{out}}(k) \, \mathcal{N}_{d1|\text{out}}(q)} + 1 \; , \tag{63}$$

and the $d1|\text{out} - d2|\text{out}$ correlator (50) takes the form

$$g_2(K,Q)_{d1|\text{out}-d2|\text{out}} = \frac{\left| S_{d1,u}^* S_{d2,u} n_U + S_{d1,d1}^* S_{d2,d1} n_{D1} + S_{d1,d2}^* S_{d2,d2}(1 + n_{D2}) \right|^2}{\mathcal{N}_{d1|\text{out}}(k) \, \mathcal{N}_{d2|\text{out}}(q)} + 1 \; , \tag{64}$$

where [from (56)]

$$\mathcal{N}_{d1|\text{out}}(k) = \left(|S_{d1,u}|^2 n_U + |S_{d1,d1}|^2 n_{D1} + |S_{d1,d2}|^2(1 + n_{D2})\right)_{\omega_{d1|\text{out}}(k)} . \tag{65}$$

Eqs. (60), (63) and (64) are particularly important because they allow us to study how an initial nonzero temperature affects the violation of the Cauchy-Schwarz inequality, $g_2 > 2$, see e.g. Eq. (51). The results in the waterfall, $\delta$-peak and flat profile configurations are presented in Figs. 6 and 7. We saw in the previous subsection that at $T = 0$ the Cauchy-Schwarz inequality is always violated along the $u|\text{out} - d2|\text{out}$ and $d1|\text{out} - d2|\text{out}$ channels for all $\omega > 0$, while it is not violated in the $u|\text{out} - d1|\text{out}$ one. In a more realistic case in which there is an initial nonzero temperature, the amount of entanglement is, as expected, reduced with respect to the $T = 0$ case. In particular, there is always a region, for small enough and large enough momenta (when the corresponding frequency is close to 0 or to $\Omega$), where $g_2 < 2$. In both the $u|\text{out} - d2|\text{out}$ and $d1|\text{out} - d2|\text{out}$ channels, however, there is always an intermediate $\omega$ region in which the Cauchy-Schwarz inequality is violated provided the temperature is not too large.

We clearly see that the most favorable case for demonstrating quantum entanglement – i.e. spontaneous Hawking radiation – corresponds to the violation of the Cauchy-Schwarz inequality along the Hawking quanta – partner ($u|\text{out} - d2|\text{out}$) channel in the waterfall configuration, which is exactly the situation which has been recently studied experimentally by Steinhauer [24]. We remark that this configuration corresponds to the case where the region of overlap of the signals in momentum space is the smallest.

We also note from Fig. 7 that, for a $\delta$-peak potential, the Cauchy-Schwarz inequality is violated up to a temperature slightly larger than the chemical potential. This has been already noticed in Ref. [32]. The signature of quantum correlation is more robust for a waterfall configuration: one sees on Fig. 6 that the Cauchy-Schwarz inequality is violated up to a temperature of order of twice the chemical potential.

## 3.3 Non adiabatic effects and *in situ* measurements.

Since the adiabatic limit described by Eqs. (33) and (34) is rather idealized, especially for small frequencies (as discussed earlier, for low lying excitations the assumption of adiabaticity is always violated), it is useful to describe how the results presented in the previous two subsections are modified by non-adiabatic effects. Concretely, instead of Eqs. (33) and (34), at late times after the opening of the trap, the Bogoliubov coefficients will behave as

$$
\begin{aligned}
\widetilde{\mathcal{U}}_{u|\text{in}} \,,\ \widetilde{\mathcal{U}}_{u|\text{out}} &\sim \alpha_u \,, & \widetilde{\mathcal{U}}_{d1|\text{in}} \,,\ \widetilde{\mathcal{U}}_{d1|\text{out}} &\sim \alpha_d \,, \\
\widetilde{\mathcal{W}}_{u|\text{in}} \,,\ \widetilde{\mathcal{W}}_{u|\text{out}} &\sim \beta_u \,, & \widetilde{\mathcal{W}}_{d1|\text{int}} \,,\ \widetilde{\mathcal{W}}_{d1|\text{out}} &\sim \beta_d \,, \\
\widetilde{\mathcal{U}}_{d2|\text{in}} \,,\ \widetilde{\mathcal{U}}_{d2|\text{out}} &\sim \beta_d \,, & \widetilde{\mathcal{W}}_{d2|\text{in}} \,,\ \widetilde{\mathcal{W}}_{d2|\text{out}} &\sim \alpha_d \,,
\end{aligned}
\tag{66}
$$

where the $\alpha$ and $\beta$ coefficients satisfy the same normalization condition (21) as the corresponding $\widetilde{\mathcal{U}}$, $\widetilde{\mathcal{W}}$ "tilde Bogoliubov coefficients" (20). Note that our approach encompasses also the situation which we denote as *in situ* below. In this situation the measurement process gives a direct access to the actual value of the tilde Bogoliubov coefficients without any evaporative process which would affect these coefficients as described in Eqs. (33) and (34) for the adiabatic limit, or possibly differently in a non adiabatic regime. In the following, we use the generic terminology "non adiabatic" for describing both the case of *in situ* measurements and of non-adiabatic modification of the situation (33), (34).

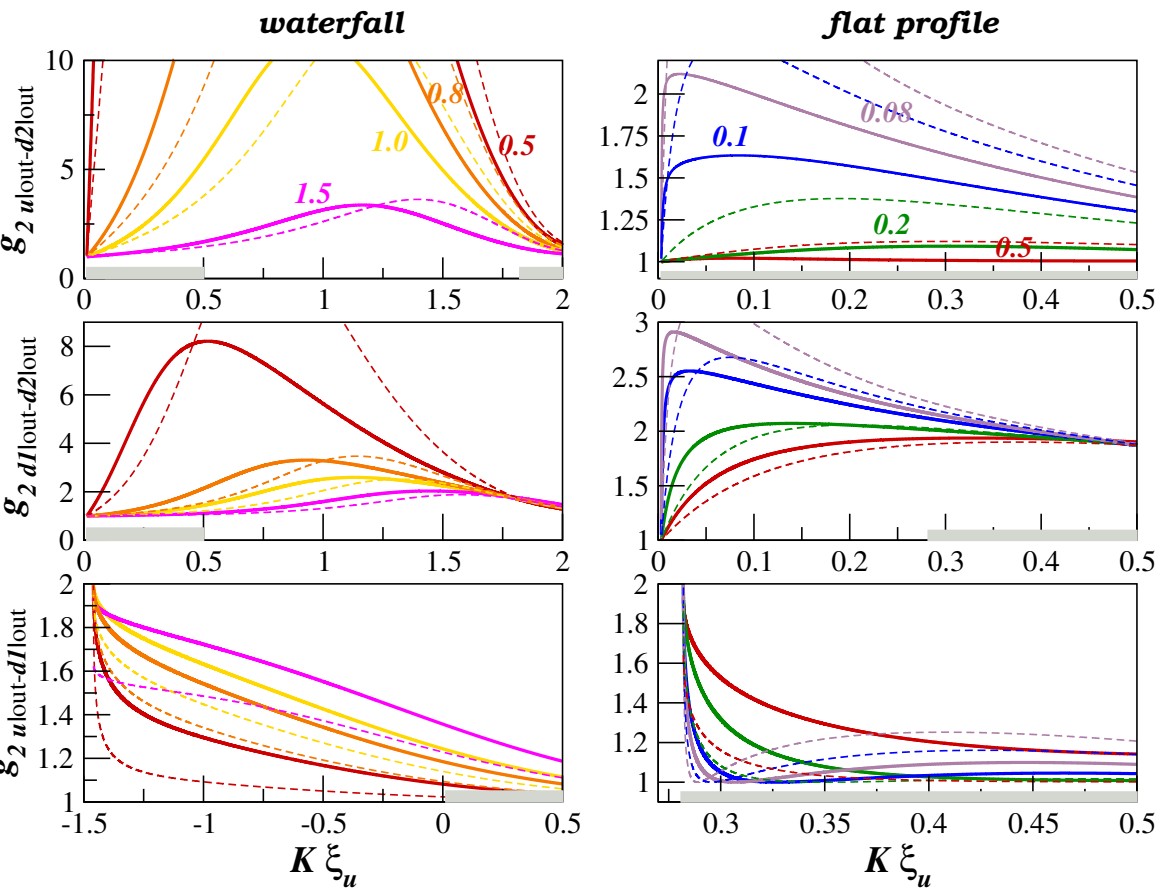

Figure 6: Normalized correlation functions in the waterfall configuration (left plots) and flat profile (right plots). $M_u = 0.5$ and $M_d = 4$ in both cases. The vertical arrangement of the different plots corresponds to the intensity of the correlation signal along the $u|\text{out} - d2|\text{out}$, $d1|\text{out} - d2|\text{out}$ and $u|\text{out} - d1|\text{out}$ correlation lines identified in Fig. 4. The Cauchy-Schwarz inequality is violated when $g_2$ is larger than 2. In each plot the different lines correspond to different temperatures, the values of which are indicated in units of $mc_u^2$. The thick solid (thin dashed) lines correspond to a situation where the system is at thermal equilibrium in the frame of the condensate (of the obstacle) before the formation of the sonic horizon. The shaded zone on the abscissa axis represent the region of overlap of the momenta of the outgoing channels, see the discussion at the end of Sec. 3.1.

To ease the presentation we have – only temporarily – considered in (66) a single set of coefficients for the upstream $u$ region and also a single one for the downstream $d$ region, but it is clear that in a rigorous analysis (presented below) each mode will have its own $\alpha$ and $\beta$ coefficients. Hence the expressions (26) and (27) remain valid, but the $\widetilde{\mathcal{U}}$ and $\widetilde{\mathcal{W}}$'s are replaced by $\alpha$ and $\beta$ coefficients, following the rules (66).

The reason for the change of notation (66) is threefold. First, the expression of the non-adiabatic coefficients maybe quite non-trivial and different from the ones of the $\widetilde{\mathcal{U}}$'s and $\widetilde{\mathcal{W}}$'s. This will occur for instance after a step of dynamical Casimir amplification where the system is artificially submitted to a rapid quench. Also, this notation allows for the possibility that,

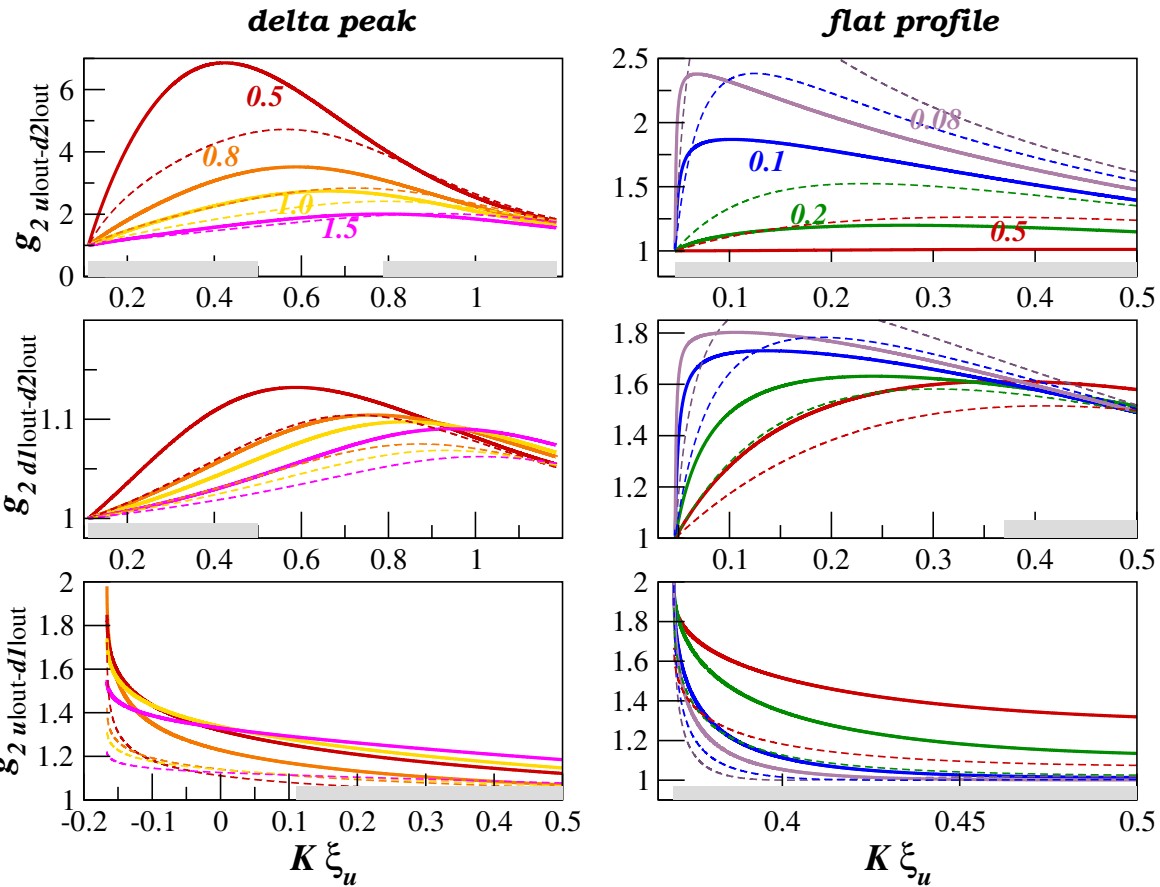

Figure 7: Same as Fig. 7 for a $\delta$-peak configuration (left plot) and a flat profile one (right plot). $M_u = 0.5$ and $M_d = 1.827$ in both cases.

during the opening of the trap, the initial "tilde Bogoliubov coefficients" get modified in a manner less trivial than (33) and (34). Second, this (momentarily) simplified notations where the mode indices are omitted permits a simple presentation of the main features of the $G_2$ function in the non-adiabatic case (see Appendix B). Finally, in the weakly non adiabatic case all the $\beta$'s are small compared with the $\alpha$'s, whereas keeping the previous $\widetilde{\mathcal{U}}$'s and $\widetilde{\mathcal{W}}$'s notations would make it difficult to identify the small terms in the expression for $G_2$.

The results for the correlator are much more complex than those presented in the previous two subsections in the adiabatic limit. We give here the general structure of the correlators, the explicit results – with also account of finite temperature – are given in Appendix B:

$$G_2(k < 0, q < 0) = \text{Diag}_{<0}\delta^2(k-q) + O(|\alpha|^4)_{<0} + O(|\alpha|^2|\beta|^2)_{<0} + O(|\beta|^4)_{<0} + (k \leftrightarrow q), \quad (67)$$

$$G_2(k > 0, q > 0) = \text{Diag}_{>0}\delta^2(k-q) + O(|\alpha|^4)_{>0} + O(|\alpha|^2|\beta|^2)_{>0} + O(|\beta|^4)_{>0} + (k \leftrightarrow q), \quad (68)$$

$$G_2(k < 0, q > 0) = A\delta^2(k+q) + O(|\alpha|^4) + O(|\alpha|^2|\beta|^2) + O(|\beta|^4). \quad (69)$$

We recall that in the adiabatic limit, corresponding to Eqs. (33) and (34) with the substitution (66), all the $\beta$'s are zero. In the more general case considered here they are not; the $\alpha$'s and $\beta$'s, with the substitution (66) – satisfy the normalization (21) – implying that $\alpha$ is bigger than its adiabatic value. We see from the results presented in Appendix B that the terms already present in the adiabatic regime are now multiplied by a factor $\alpha^4$. In particular, the finite temperature adiabatic terms of (57), (58) and (59) are now given by the $\alpha^4$ diagonal terms given in Eqs. (127) and (131) and by the off-diagonal terms $O(|\alpha|^4)_{<0}$ in Eq. (128), $O(|\alpha|^4)_{>0}$ in Eq. (132), and $O(|\alpha|^4)$ in Eq. (136). If we consider the weakly non adiabatic regime, this means that they are now larger than the corresponding adiabatic value. New sub-leading terms appear, of order $\alpha^2 \beta^2$ terms (among which an antidiagonal term) and also higher order $\beta^4$ contribution. This results in a very complicated pattern.

### 3.3.1 Violation of the Cauchy-Schwarz inequality along the Hawking quantum-partner correlation lines

Having derived the structure of the correlation lines, we now turn to the intensity of the correlation signal, and to the possible violation of the Cauchy-Schwarz inequality in the non adiabatic regime. We shall restrict our attention to the study of the incidence of non-adiabaticity on the Hawking quantum - partner $u|$out-$d2|$out correlator; this will bring pieces of information valid for all the other correlation lines. The results will be given first at $T = 0$[4], then at finite temperature for completeness.

• From the results in Appendix B, in the case where both $k$ and $q$ are negative, the zero temperature contribution of the $u|$out–$d2|$out modes to $G_2$ reads [see Eq. (128)]

$$G_2(k,q)_{u|\text{out}-d2|\text{out}} \longleftarrow |\alpha_{u|\text{out}}|^2 |\alpha_{d2|\text{out}}|^2 |S_{u,d2}|^2 |S_{d2,d2}|^2 \delta^2(\omega_{u|\text{out}}(k) - \omega_{d2|\text{out}}(-q)), \quad (70)$$

The arrow in this equation indicates that its right hand side is not the sole contribution to the $u|$out–$d2|$out correlation signal: not only should it be supplemented by a contribution in which the roles of $k$ and $q$ are exchanged, but also new non-adiabatic terms arise (see below). The contribution (70) corresponds to the signal which already exists in the adiabatic case, whose intensity is here modified by the $\alpha$ coefficients. Note that, contrarily to the schematic presentation of Appendix B, we consider here the most general case, and have explicitly written the mode-dependence of the $\alpha$ coefficients. This will remain the case in the rest of the section.

In the negative $k$ sector, the zero temperature contributions of the $u|$out and $d2|$out modes to $\mathcal{N}(k)$ read

$$\begin{aligned}
\mathcal{N}_{u|\text{out}}(k < 0) &= |\alpha_{u|\text{out}}|^2 |S_{u,d2}|^2 |_{\omega_{u|\text{out}}(k)} \ , \\
\mathcal{N}_{d2|\text{out}}(k < 0) &= |\alpha_{d2|\text{out}}|^2 (|S_{d2,d2}|^2 - 1)|_{\omega_{d2|\text{out}}(-k)} \ .
\end{aligned} \quad (71)$$

In the calculation of $g_2(K,Q)_{u|\text{out}-d2|\text{out}}$ the $\alpha$ coefficients of Eqs. (70) and (71) factorize out and we get for the normalized correlator the same result as in zero temperature adiabatic case, Eq. (47), namely

$$g_2(K,Q)_{u|\text{out}-d2|\text{out}} \longleftarrow \frac{2|S_{d2,d2}|^2 - 1}{|S_{d2,d2}|^2 - 1} \geq 2 \ . \quad (72)$$

---

[4]A similar analysis in the case of the density correlator has been presented in [49].

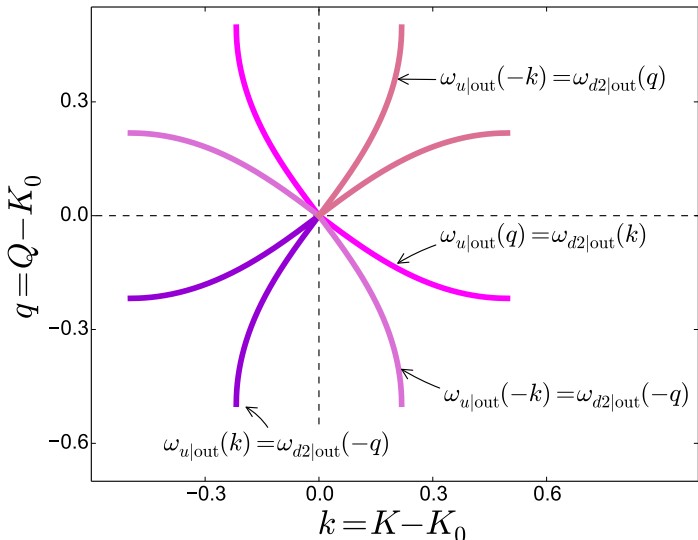

Figure 8: Momentum space correlation lines for the $u|$out-$d2|$out signal in a non-adiabatic case for a flat profile configuration with $M_u = 0.5$ and $M_d = 4$. The correlation line corresponding to $\omega_{u|\text{out}}(k) = \omega_{d2|\text{out}}(-q)$ is already present in the adiabatic regime (see the right plot of Fig. 4). The lines of identical colors are obtained one from the other by an exchange of $k$ and $q$. The same pattern arises *in situ*. The momenta are expressed in units of $\xi_u^{-1}$.

The same factorization of the $\alpha$ coefficients occurs also at finite temperature, and the adiabatic expression (60) is thus also valid in the non-adiabatic regime. We stress that this important signal is thus quite robust, not being affected by possible non adiabatic effects, and the violation of Cauchy-Schwarz inequality is also observable from *in situ* measurements.

From the results presented in Appendix B one sees that in the non-adiabatic regime three more couples of $u|$out–$d2|$out correlation lines appear with respect to the adiabatic situation. They correspond to the conditions

$$
\begin{aligned}
\omega_{d2|\text{out}}(k) &= \omega_{u|\text{out}}(-q) \quad \text{for} \quad k > 0 \text{ and } q > 0 \,, \\
\omega_{d2|\text{out}}(q) &= \omega_{u|\text{out}}(k) \qquad \text{for} \quad k < 0 \text{ and } q > 0 \,, \\
\omega_{d2|\text{out}}(-q) &= \omega_{u|\text{out}}(-k) \quad \text{for} \quad k > 0 \text{ and } q < 0 \,,
\end{aligned}
\tag{73}
$$

and to the same conditions in which the roles of $k$ and $q$ are exchanged. All these correlation lines are displayed in Fig. 8, together with the ones which already exist in the adiabatic case[5]. We now evaluate the intensity of the signal corresponding to each of these lines.

• In the $k$ and $q > 0$ sector, we get, at $T = 0$ [cf. Eq. (134)]:

$$
G_2(k,q)_{u|\text{out}-d2|\text{out}} \longleftarrow |\beta_{u|\text{out}}|^2 |\beta_{d2|\text{out}}|^2 |S_{u,d2}|^2 |S_{d2,d2}|^2 \delta^2(\omega_{d2|\text{out}}(k) - \omega_{u|\text{out}}(-q))
\tag{74}
$$

and

$$
\begin{aligned}
\mathcal{N}_{u|\text{out}}(k > 0) &= |\beta_{u|\text{out}}|^2 (1 + |S_{u,d2}|^2)|_{\omega_{u|\text{out}}(-k)} \,, \\
\mathcal{N}_{d2|\text{out}}(k > 0) &= |\beta_{d2|\text{out}}|^2 |S_{d2,d2}|^2|_{\omega_{d2|\text{out}}(k)} \,,
\end{aligned}
\tag{75}
$$

---

[5]For legibility we only show in Fig. 8 the $u|$out–$d2|$out correlation lines. In the non-adiabatic case there are many more similar lines, corresponding to the correlations identified in Appendix B.

leading to the following contribution to the normalized correlator:

$$g_2(K,Q)_{u|\text{out}-d2|\text{out}} \longleftarrow 2 - \frac{1}{1+|S_{u,d2}|^2} \le 2 \ . \tag{76}$$

As was previously the case for the $\alpha$ coefficients, the $\beta$'s here also factorize out. A similar factorization will occur in all the subsequent cases considered in this section. At finite temperature, expression (76) becomes

$$g_2(K,Q)_{u|\text{out}-d2|\text{out}} \longleftarrow \frac{\left| S_{u,u}^* S_{d2,u} n_U + S_{u,d1}^* S_{d2,d1} n_{D1} + S_{u,d2}^* S_{d2,d2}(1+n_{D2}) \right|^2}{\mathscr{N}_{u|\text{out}}(k>0)\,\mathscr{N}_{d2|\text{out}}(q>0)} + 1 \ , \tag{77}$$

where here

$$\begin{aligned}
\mathscr{N}_{u|\text{out}}(k>0) &\equiv \frac{1}{|\beta_{u|\text{out}}|^2}\mathcal{N}_{u|\text{out}}(k>0) \\
&= \left(|S_{u,u}|^2(1+n_U) + |S_{u,d1}|^2(1+n_{D1}) + |S_{u,d2}|^2 n_{D2}\right)_{\omega_{u|\text{out}}(-k)} \ , \\
\mathscr{N}_{d2|\text{out}}(q>0) &\equiv \frac{1}{|\beta_{d2|\text{out}}|^2}\mathcal{N}_{d2|\text{out}}(q>0) \\
&= \left(|S_{d2,u}|^2 n_U + |S_{d2,d1}|^2 n_{D1} + |S_{d2,d2}|^2(1+n_{D2})\right)_{\omega_{d2|\text{out}}(q)} \ .
\end{aligned} \tag{78}$$

The important information here is that, even at zero temperature, the new $u|\text{out}$–$d2|\text{out}$ correlation line which appears in the $k$ and $q>0$ sector due to non-adiabatic effects is not associated to a non separable signal [cf. Eq. (76)]. As expected –and can be verified from expression (77)– thermal effects do not modify this situation. As we will now see, this conclusion remains also valid for all the other correlation lines which where not present in the adiabatic regime.

• For $k<0$ and $q>0$, one of the contributions to $G_2(k,q)_{u|\text{out}-d2|\text{out}}$ reads [cf. Eq. (137)]

$$G_2(k,q)_{u|\text{out}-d2|\text{out}} \longleftarrow |\alpha_{u|\text{out}}|^2|\beta_{d2|\text{out}}|^2|S_{u,d2}|^2|S_{d2,d2}|^2\delta^2(\omega_{d2|\text{out}}(q) - \omega_{u|\text{out}}(k)) \ , \tag{79}$$

and at $T=0$ this corresponds to a normalized correlation signal:

$$g_2(K,Q)_{u|\text{out}-d2|\text{out}} \longleftarrow 2 \ . \tag{80}$$

At finite temperature this contribution modifies to:

$$g_2(K,Q)_{u|\text{out}-d2|\text{out}} \longleftarrow \frac{\left| S_{u,u}^* S_{d2,u} n_U + S_{u,d1}^* S_{d2,d1} n_{D1} + S_{u,d2}^* S_{d2,d2}(1+n_{D2}) \right|^2}{\mathscr{N}_{u|\text{out}}(k<0)\,\mathscr{N}_{d2|\text{out}}(q>0)} + 1 \ , \tag{81}$$

where

$$\begin{aligned}
\mathscr{N}_{u|\text{out}}(k<0) &\equiv \frac{1}{|\alpha_{u|\text{out}}|^2}\mathcal{N}_{u|\text{out}}(k<0) \\
&= \left(|S_{u,u}|^2 n_U + |S_{u,d1}|^2 n_{D1} + |S_{u,d2}|^2(1+n_{D2})\right)_{\omega_{u|\text{out}}(-k)} \ .
\end{aligned} \tag{82}$$

• For $k>0$ and $q<0$ another contribution to $G_2(k,q)_{u|\text{out}-d2|\text{out}}$ reads [cf. Eq. (137)]

$$G_2(k,q)_{u|\text{out}-d2|\text{out}} \longleftarrow |\alpha_{d2|\text{out}}|^2|\beta_{u|\text{out}}|^2|S_{u,d2}|^2|S_{d2,d2}|^2\delta^2(\omega_{d2|\text{out}}(-q) - \omega_{u|\text{out}}(-k)) \ , \tag{83}$$

giving at $T = 0$

$$g_2(K, Q)_{u|\text{out}-d2|\text{out}} \longleftarrow 2 - \frac{|S_{d1,d2}|^2}{(|S_{d2,d2}|^2 - 1)(1 + |S_{u,d2}|^2)} \leq 2 . \tag{84}$$

This form of writing the result has been obtained by using the pseudo-unitarity of the $S$ matrix. It makes clear that no violation of the Cauchy-Schwarz inequality occurs along the correlation line considered here. At finite temperature one obtains

$$g_2(K, Q)_{u|\text{out}-d2|\text{out}} \longleftarrow \frac{\left| S_{u,u}^* S_{d2,u} n_U + S_{u,d1}^* S_{d2,d1} n_{D1} + S_{u,d2}^* S_{d2,d2}(1 + n_{D2}) \right|^2}{\mathcal{N}_{u|\text{out}}(k > 0) \, \mathcal{N}_{d2|\text{out}}(q < 0)} + 1 , \tag{85}$$

where here

$$\begin{aligned} \mathcal{N}_{d2|\text{out}}(q < 0) &\equiv \frac{1}{|\alpha_{d2|\text{out}}|^2} \mathcal{N}_{d2|\text{out}}(q < 0) \\ &= \left( |S_{d2,u}|^2(1 + n_U) + |S_{d2,d1}|^2(1 + n_{D1}) + |S_{d2,d2}|^2 n_{D2} \right)_{\omega_{d2|\text{out}}(-q)} . \end{aligned} \tag{86}$$

To sum up, we recall that the above study is a partial focus on a subpart of the whole correlation pattern, concerning only the most important Hawking quantum-partner signal (in our terminology, the $u|\text{out} - d2|\text{out}$ signal). We used it to demonstrate that the most interesting correlation is the one already present in the $T = 0$ adiabatic case: the Cauchy-Schwarz inequality can only be violated along this line. The same is true for the $d1|\text{out} - d2|\text{out}$ signal.

## 4 Momentum correlations in the absence of sonic horizon

In this section we consider a configuration where the upstream and the downstream regions are both subsonic. In this case there is no horizon, but one can still be in a configuration where the upstream and downstream non linear coefficients are different (as in the flat profile configuration), also the system can be affected by the presence of an external potential (as in the waterfall and delta peak configuration). If this external potential is localized (i.e., tends rapidly enough to zero at infinity), and if the nonlinear coefficient keeps the same value in all the system, then the type of flow considered is rather simple. More precisely, as demonstrated in Appendix C, the upstream flow velocity and density at $-\infty$ are the same as the flow velocity and density at $+\infty$: $V_u = V_d$ and $n_u = n_d$. As a result, the general formulas given in the present section simplify, this is explained in Appendix C.

In the situation we consider in the present section, the $d2$ negative-norm mode disappears since the downstream region is subsonic: there is now a single downstream mode which we simply denote by "$d$". The obstacle is thus characterized by a $S$-matrix which is $2 \times 2$ and unitary:

$$S = \begin{pmatrix} S_{u,u} & S_{u,d} \\ S_{d,u} & S_{d,d} \end{pmatrix} \quad \text{with} \quad SS^\dagger = \mathbb{1} . \tag{87}$$

In this case, instead of Eq. (8) one has

$$\hat{\psi}(x) = e^{iK_\alpha x} \int_0^\infty \frac{d\omega}{\sqrt{2\pi}} \sum_{L \in \{U, D\}} \left[ \bar{u}_L(x, \omega) \hat{b}_L(\omega) + \bar{w}_L^*(x, \omega) \hat{b}_L^\dagger(\omega) \right] , \tag{88}$$

where $K_\alpha = mV_\alpha/\hbar$ as in (8), $V_\alpha$ being the value of the upstream ($\alpha = u$) or downstream ($\alpha = d$) asymptotic flow velocity.

We perform the usual (fake) Fourier transform on $\hat{\psi}$, using the rules R1 and R2 of section (2.4.1). Collecting separately the $k < 0$ and $k > 0$ contributions we get

$$
\begin{aligned}
\hat{\psi}(k < 0) &= \widetilde{\mathcal{U}}_{u|\text{out}}\left(S_{u,u}\,\hat{b}_U + S_{u,d}\,\hat{b}_D\right)_{\omega_{u|\text{out}}(k)} + \widetilde{\mathcal{W}}^*_{d|\text{out}}\left(S^*_{d,u}\,\hat{b}^\dagger_U + S^*_{d,d}\,\hat{b}^\dagger_D\right)_{\omega_{d|\text{out}}(-k)} \\
&\quad + \left.\widetilde{\mathcal{U}}_{d|\text{in}}\,\hat{b}_D\right|_{\omega_{d|\text{in}}(k)} + \left.\widetilde{\mathcal{W}}^*_{u|\text{in}}\,\hat{b}^\dagger_U\right|_{\omega_{u|\text{in}}(-k)} \,,
\end{aligned}
\tag{89}
$$

$$
\begin{aligned}
\hat{\psi}(k > 0) &= \widetilde{\mathcal{U}}_{d|\text{out}}\left(S_{d,u}\,\hat{b}_U + S_{d,d}\,\hat{b}_D\right)_{\omega_{d|\text{out}}(k)} + \widetilde{\mathcal{W}}^*_{u|\text{out}}\left(S^*_{u,u}\,\hat{b}^\dagger_U + S^*_{u,d}\,\hat{b}^\dagger_D\right)_{\omega_{u|\text{out}}(-k)} \\
&\quad + \left.\widetilde{\mathcal{U}}_{u|\text{in}}\,\hat{b}_U\right|_{\omega_{u|\text{in}}(k)} + \left.\widetilde{\mathcal{W}}^*_{d|\text{in}}\,\hat{b}^\dagger_D\right|_{\omega_{d|\text{in}}(-k)} \,.
\end{aligned}
$$

This expression corresponds to Eqs. (26) and (27) in which the negative norm $d2$ mode has been suppressed. From it one can compute the density distribution in momentum space and also the momentum correlation function. In particular, in the adiabatic limit one gets $\langle \hat{N}(k < 0) \rangle = \mathcal{N}(k) \times \delta(k - k)$ where

$$
\mathcal{N}(k < 0) = \left(|S_{u,u}|^2 n_U + |S_{u,d}|^2 n_D\right)_{\omega_{u|\text{out}}(k)} + \left. n_D \right|_{\omega_{d|\text{in}}(k)} \,,
\tag{90}
$$

and

$$
\mathcal{N}(k > 0) = \left(|S_{d,d}|^2 n_D + |S_{d,u}|^2 n_U\right)_{\omega_{d|\text{out}}(k)} + \left. n_U \right|_{\omega_{u|\text{in}}(k)} \,.
\tag{91}
$$

Since one is in a situation where the flow is everywhere subsonic, one can define a *bona fide* temperature state where, for all the modes, the occupation number of a state of energy $\omega$ is $n_U = n_D = n_{\text{th}}(\omega)$. In the idealized case of perfect transmission one has $S_{u,u} = 0 = S_{d,d}$ and $S_{u,d} = 1 = S_{d,u}$, then Eqs. (90) and (91) reduce to

$$
\mathcal{N}(k) = 2\, n_{\text{th}}(\omega(k)) \,.
\tag{92}
$$

The factor 2 is spurious. It comes from the fact that one does two Fourier transforms, one upstream and one downstream, and that there is a kind of built-in double counting in this approach. The problem is suppressed if one considers upstream and downstream windowed Fourier transforms, as done in Appendix A. In this case, in the absence of the obstacle, instead of Eqs. (90) and (91) one gets

$$
\langle \hat{N}(k) \rangle = n_{\text{th}}(\omega(k)) \left(\frac{\sigma_u \Lambda_u^2}{\sqrt{8\pi}} + \frac{\sigma_d \Lambda_d^2}{\sqrt{8\pi}}\right) \,.
\tag{93}
$$

How the correct treatment of the windowed upstream and downstream Fourier transforms leads to the precise form of the two terms in the big parenthesis of the r.h.s. of (93) is explained in Appendix A.

From the present analysis one is thus led to define finite efficiencies of the upstream and downstream particle detectors. A finite efficiency corresponds to a measurement process in which a fraction of particles are missed in the detection of the momentum signal. This is described theoretically by the normalization of the windowed Fourier transforms (16): defining the efficiencies as $\lambda_\alpha = \sigma_\alpha \Lambda_\alpha^2/\sqrt{2\pi}$ ($\alpha = u$ or $d$) with $\lambda_\alpha \in [0, 1]$ one casts formula (93) under the form $\langle \hat{N}(k) \rangle = \frac{1}{2}(\lambda_u + \lambda_d) n_{\text{th}}(\omega(k))$. So, when both efficiencies are zero, there is no particle

detected and no momentum signal, and when, on the contrary, the detection efficiencies are both unity one gets in the absence of the obstacle $\langle \hat{N}(k) \rangle = n_{\text{th}}(\omega(k))$, as it should be.

Note that the double counting which has been easily identified in formula (92) is present in all the previous formulas of main text (Eqs. (53) to (56) and also Eqs. (57) to (59)). It can be cured in the same simple way. For instance, considering perfect detectors, one should have added a factor $1/\sqrt{2}$ in the definition of rules R1 and R2 of section (2.4.1) for the schematic Fourier transforms. We did not do that for avoiding an overall multiplicative factor in all the formulas, and also because, as demonstrated in Appendix A this double counting (and also the finite efficiency of the detector) disappears in the formulas for the normalized correlation signal $g_2$ which is our main interest in the present work.

Finally, we give the expression for the momentum correlator, for simplicity in the adiabatic regime

$$
\begin{aligned}
G_2(k < 0, q < 0) &= \mathcal{N}^2(k < 0)\delta^2(k - q) \\
&+ \left( |\widetilde{\mathcal{U}}_{u|\text{out}}|^2 |\widetilde{\mathcal{U}}_{d|\text{out}}|^2 |S_{u,d}|^2 n_D^2 \delta^2(\omega_{u|\text{out}}(k) - \omega_{d|\text{in}}(q)) + (k \leftrightarrow q) \right),
\end{aligned}
\tag{94}
$$

$$
\begin{aligned}
G_2(k > 0, q > 0) &= \mathcal{N}^2(k > 0)\delta^2(k - q) \\
&+ \left( |\widetilde{\mathcal{U}}_{u|\text{in}}|^2 |\widetilde{\mathcal{U}}_{d|\text{out}}|^2 |S_{d,u}|^2 n_U^2 \delta^2(\omega_{d|\text{out}}(k) - \omega_{u|\text{in}}(q)) + (k \leftrightarrow q) \right),
\end{aligned}
\tag{95}
$$

$$
\begin{aligned}
G_2(k < 0, q > 0) &= |\widetilde{\mathcal{U}}_{u|\text{out}}|^2 |\widetilde{\mathcal{U}}_{d|\text{out}}|^2 \left| S_{u,u}^* S_{d,u} n_U + S_{u,d}^* S_{d,d} n_D \right|^2 \delta^2(\omega_{u|\text{out}}(k) - \omega_{d|\text{out}}(q)) \\
&+ |\widetilde{\mathcal{U}}_{u|\text{out}}|^2 |\widetilde{\mathcal{U}}_{u|\text{in}}|^2 |S_{u,u}|^2 n_U^2 \delta^2(\omega_{u|\text{out}}(k) - \omega_{u|\text{in}}(q)) \\
&+ |\widetilde{\mathcal{U}}_{d|\text{out}}|^2 |\widetilde{\mathcal{U}}_{d|\text{in}}|^2 |S_{d,d}|^2 n_D^2 \delta^2(\omega_{d|\text{in}}(k) - \omega_{d|\text{out}}(q)) .
\end{aligned}
\tag{96}
$$

These expressions can be further simplified by considering, for this configuration which is everywhere subsonic, a common occupation number $n_{\text{th}}(\omega)$ of a state of energy $\omega$. As expected the corresponding expressions in the black hole case, Eqs. (57-59), reduce to the above in the absence of the negative norm $d2$ modes. Also, unlike in the black hole case all contributions disappear in the case where the initial state is the vacuum.

# 5  Limitations of the theoretical description

In this section we properly define the domain of applicability of our approach. A first limitation concerns the low density regime. As well known, in one dimension phase fluctuations prevent a true Bose-Einstein condensation. As a result, a description based on the simple separation (1) between a classical field and a small quantum correction does not allow to properly estimate the large $x$ behavior of the one body density matrix $\langle \Psi^\dagger(x)\Psi(0) \rangle$ of a homogeneous 1D system (see, e.g., Ref. [50]) and, at low density, phase fluctuations blur the sharp correlations of Figs. 4 and 5, cf. Refs. [51] and [52]. We nonetheless argue that (1) it is still useful for understanding the qualitative behavior of some observables important for analyzing correlations in the system: for instance (1) yields the correct two-body correlation $\langle \hat{\Psi}^\dagger(0)\hat{\Psi}^\dagger(x)\hat{\Psi}(x)\hat{\Psi}(0) \rangle$ of a homogeneous system [53].

The relevance of (1) depends on the characteristic length involved in the spatial correlation considered. If the correlation characteristic length is smaller than the phase coherence length

$L_\phi = \xi \exp\left[\pi\sqrt{\frac{\hbar n}{2ma\omega_\perp}}\right]$ then (1) may be used. This is what happens for the two-body correlation : this quantity is non trivial (i.e., different from the square $n^2$ of the linear density $n = \langle\hat\Psi^\dagger(0)\hat\Psi(0)\rangle$) only in a range of distances $x < \xi$, typically much smaller than $L_\phi$. More precisely, $L_\phi$ is exponentially larger than $\xi$, and the separation (1) is thus valid, when

$$\left(\frac{a}{a_\perp}\right)^2 \ll n\,a\,, \tag{97}$$

where $a_\perp = \sqrt{\hbar/m\omega_\perp}$ is the transverse harmonic oscillator length.

In the large density limit the 1D description also fails, not because of lack of BEC as just discussed, but because the transverse degrees of freedom of the system are not completely frozen. A realistic 3D black-hole configuration has been first considered in Ref. [54], with also account for 3 body losses. In the present discussion we focus on the treatment of transverse excitations. Assuming that Bose condensation is total, but not disregarding the transverse degrees of freedom, one considers the dynamics of the system as described by a Gross-Pitaevskii equation for the classical field $\Psi_0(\vec r, t)$:

$$i\hbar\partial_t\Psi_0 = \left(-\frac{\hbar^2\vec\nabla^2}{2m} + V_\perp(\vec r_\perp) + g\,|\Psi_0|^2\right)\Psi_0\,, \tag{98}$$

where $\vec r = x\,\vec e_x + \vec r_\perp$, $\vec r_\perp$ denoting the transverse coordinate, $V_\perp(\vec r_\perp) = \frac{1}{2}m\omega_\perp^2 r_\perp^2$ being the transverse potential and $g = 4\pi\hbar^2 a/m$. In (98) the normalization is chosen so that $\rho_0(\vec r, t) = |\Psi_0(\vec r, t)|^2$ is the density of particles. At equilibrium, in the so-called Thomas-Fermi limit [55], the Laplacian term in (98) can be omitted and the density has a cylindrical symmetry with

$$\rho_0(r_\perp) = \begin{cases} \frac{1}{g}\left[\mu - V_\perp(\vec r_\perp)\right] & \text{if } \mu \geq V_\perp(\vec r_\perp)\,, \\ 0 & \text{elsewhere}\,. \end{cases} \tag{99}$$

Here $\mu$ is the chemical potential fixed by the normalization: $\int d^2r_\perp \rho_0(r_\perp) = n$; $\mu = 2\hbar\omega_\perp\sqrt{a\,n}$ [56, 57]. The Thomas-Fermi approximation holds in the large density limit $a\,n \gg 1$ [28, 58]. In this limit, which has been denoted as "3D cigar" in Ref. [59], the classical field description is accurate, that is, the quantum fluctuations around $\Psi_0$ are small.

In the cylindrical geometry we consider here, the excitation spectrum has several branches corresponding to density fluctuations of the form $\delta\rho_0(\vec r, t) = \delta\rho(r_\perp)e^{im\theta}e^{i(qx-\omega t)}$, where $\theta$ is a polar angle in the transverse plane. For each branch the lowest state is obtained for $m = 0$ and $q = 0$ and its energy reads $\hbar\omega_n = \hbar\omega_\perp\sqrt{2n(n+1)}$, with $n \in \mathbb{N}$. Taking into account possible longitudinal excitations one gets in the long wave-length limit [56, 57]

$$\omega_0^2(q) = c_{\mathrm{TF}}^2\,q^2\left(1 - \frac{1}{48}(qR_\perp)^2 + \dots\right)\,, \tag{100}$$

$$\omega_{n\geq 1}^2(q) = 2n(n+1)\,\omega_\perp^2 + c_{\mathrm{TF}}^2\,q^2 + \dots\,, \tag{101}$$

where $c_{\mathrm{TF}} = \sqrt{\mu/2\,m}$ is the sound velocity in the Thomas-Fermi limit (which has been measured by the MIT group [60]) and $R_\perp = 2\,c_{\mathrm{TF}}/\omega_\perp$ is the transverse extension of the condensate (in the same limit). Eqs. (100) and (101) describe a lower mode with sonic-like dispersion relation and gaped transverse excited states which behave quadratically at low $q$. Note that the low $q$ expansion displayed in Eq. (100) does not correspond to what is expected from the

usual Bogoliubov dispersion relation (6). This is a hint that the longitudinal dynamics of the system is modified in the Thomas-Fermi limit. Of course, the hydrodynamic result (100) is limited to the region $q \ll R_\perp^{-1}$ and cannot provide a reliable description of the whole excitation spectrum. But the departure from the usual Bogoliubov dispersion is confirmed by numerical solutions of Bogoliubov-de Gennes equations [61, 62] which are valid for the whole range of wave vectors and for a range of densities larger than those based on the Thomas-Fermi approximation. These computations show that when increasing the linear density starting from a value $n \sim a^{-1}$ (i.e., when one goes deeper in the Thomas-Fermi regime) the dispersion relation develops a plateau in the region $q \sim 1/R_\perp$. This is interpreted as a tendency of the excitations to explore the radial parts of the condensate where the density is lower and where the local sound velocity accordingly decreases. This effect is not taken into account in the theoretical analysis presented in the main text where we work in a regime which has been denoted as "1D mean field" in Ref. [59]. At zero temperature this corresponds to the regime where the condition (97) is supplemented by

$$n\,a \ll 1 \,. \tag{102}$$

In this regime $\mu = 2\hbar\omega_\perp an$ [26] which is much smaller that the energy $2\,\hbar\omega_\perp$ of the first transverse excited state[6], one can thus safely neglect transverse excitations and the transverse density profile is not of the type (99), but has rather a Gaussian shape.

It is interesting to evaluate the actual range of parameters corresponding to the fulfillment of conditions (97) and (102), which is the regime of validity of our approach. For a transverse trap of frequency of 1 kHz, one gets for $^{23}$Na $(a_\perp/a)^2 = 1.7 \times 10^{-5}$, for $^{87}$Rb $(a_\perp/a)^2 = 2.6 \times 10^{-4}$ and for He* $(a_\perp/a)^2 = 2.2 \times 10^{-5}$ [64]. Hence the domain of validity of the 1D mean field approximation used in the present work ranges over four orders of magnitudes in density.

In present time experiments, when the 1D mean field regime fails, this is mostly due to the fact that the the linear density is large, and in this case (102) may be violated. Then, the transverse density profile has the Thomas-Fermi shape (99). It is thus of interest to briefly and qualitatively discuss the features appearing in momentum correlators in acoustic black holes due to the transverse modes (101)[7].

A first remark is in order here: the new transverse modes are typically not coupled to the modes studied in the present work. The reason for this is that the potential $V(x)$ used to implement the sonic horizon does not couple modes with different transverse quantum numbers. Only small imperfections and nonlinear effect would induce such a coupling, and the results presented in the present work would remain almost unaffected. If the dispersion relation were limited to expression (101), i.e., were of the Klein-Gordon type, new outgoing modes would appear which would be populated by the time dependent formation process of the horizon, as in the gravitational context. In the present case however, the transverse dispersion relation (101) encompasses terms of higher order in $q$, and, as a result, new transverse incoming modes appear in the supersonic region, of the $d2|$in type, as illustrated in Fig. 9. As a result of the existence of these new incoming modes, the Hawking radiation process would occur also in the transverse sector even in the stationary context, and consequently new correlations lines appear which should add to the ones studied in the main text. However, in the regime (102) they should correspond to a very weak signal.

---

[6]The value $2\,\hbar\omega_\perp$ is the same as $\hbar\omega_{n=1}(q=0)$ in (101) : it is model independent as a result of a scaling property of the Gross-Pitaevskii equation in two dimensions [63].

[7]Transverse modes can be incorporated in our analysis following the approach of Ref. [68].

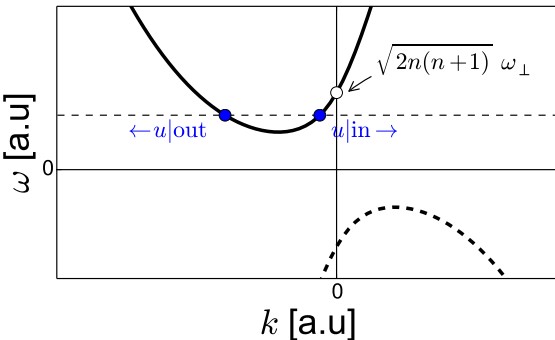 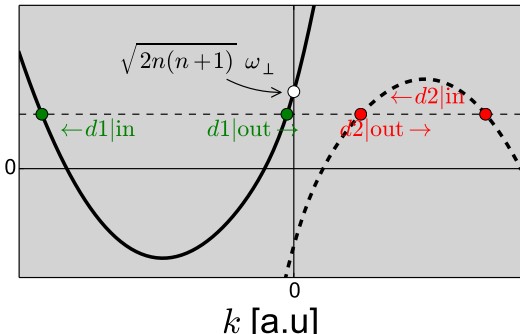

Figure 9: Sketch of the typical dispersion relations of a transverse mode in the subsonic (left plot) and supersonic (right plot) case. The horizontal dashed line is fixed by the chosen value of $\omega$.

## 6    Conclusions

In this work we have investigated in detail the two-body momentum correlation in a quasi 1D BEC in the presence of a sonic horizon. Our modeling of the measurement process shows that the measurements have to be performed with some care: (1) the spatial windows selected for the Fourier analysis must be chosen carefully in order not to damp the expected signal (see Appendix A and also Ref. [34]) and (2) a separation of the upstream and downstream signals in the detection scheme favors the highlighting of quantum non separability (end of Sec. 3.1). Once the appropriate requirements are met, the normalized correlator (29) appears to be a robust quantity, making it possible to test quantum entanglement in a rich variety of situations, namely, *in situ*, or after artificially inducing a quench in the system or at the end of an adiabatic expansion after opening of the trap. Among the possible implementations of a sonic horizon we have studied, the largest quantum correlation signal, observed between the Hawking quantum and its partner, is realized in the so-called "waterfall configuration". In this configuration the Cauchy-Schwarz inequality is violated up to a temperature larger than the chemical potential and therefore should be experimentally testable in a finite temperature setting [the violation is still present at $T = 1.5 \times mc_u^2$, see Fig. 6, upper left panel].

**Acknowledgements**    Inspiring exchanges with P.-É. Larré are gratefully acknowledged. We also thank D. Boiron, M. Isoard, G. Martone, C. Westbrook and P. Ziń for fruitful discussions.

**Funding information**    This work was supported by the French ANR under grant n° ANR-15-CE30-0017 (Haralab project) and by the Spanish Mineco grant FIS2014-57387-C3-1-P, the Generalitat Valenciana project SEJI/2017/042 and the Severo Ochoa Excellence Center Project SEV-2014-0398.

# A   Rigorous local Fourier transform

In this appendix we present the precise form of the Fourier transforms performed by using the window functions which, in the upstream region, are of the form (16).

We first note that the Fourier transform of the window function is:

$$
\Pi_u(K) = \int_{\mathbb{R}} \frac{\mathrm{d}x}{\sqrt{2\pi}} \, \mathrm{e}^{-\mathrm{i}Kx} \, \Pi_u(x) = \frac{\Lambda_u \sigma_u}{\sqrt{2}} \, \mathrm{e}^{-\frac{1}{4}K^2\sigma_u^2 - \mathrm{i}KX_u} = \sqrt{2\pi} \, \Lambda_u \, \mathrm{e}^{-\mathrm{i}KX_u} \delta_u^{(1)}(K) \,, \quad (103)
$$

where $\delta_u^{(1)}(K) = (\sigma_u/2\sqrt{\pi}) \exp\{-K^2\sigma_u^2/4\}$ is an approximation of the Dirac distribution (tending towards the $\delta$ function when $\sigma_u \to \infty$). One also has

$$
|\Pi_u(K)|^2 = \sqrt{\frac{\pi}{2}} \, \sigma_u \, \Lambda_u^2 \, \delta_u^{(2)}(K) \,, \tag{104}
$$

where $\delta_u^{(2)}(K) = (\sigma_u/\sqrt{2\pi}) \exp\{-K^2\sigma_u^2/2\}$ is another approximation of the Dirac $\delta$-distribution defined by (104) and verifying $\delta_u^{(2)}(0) = \sigma_u/\sqrt{2\pi}$. So, instead of the approximate formula (18) one gets

$$
\begin{aligned}
\hat{\psi}_u(K) = \int_0^\infty \frac{\mathrm{d}\omega}{\sqrt{2\pi}} \Big\{ & \Pi_u(K - K_u - q_{u|\text{out}})\mathcal{U}_{u|\text{out}}(S_{u,u}\hat{b}_U + S_{u,d1}\hat{b}_{D1} + S_{u,d2}\hat{b}_{D2}^\dagger) \\
& + \Pi_u(K - K_u + q_{u|\text{out}})\mathcal{W}_{u|\text{out}}^*(S_{u,u}^*\hat{b}_U^\dagger + S_{u,d1}^*\hat{b}_{D1}^\dagger + S_{u,d2}^*\hat{b}_{D2}) \\
& + \Pi_u(K - K_u + q_{u|\text{in}})\mathcal{W}_{u|\text{in}}^*\hat{b}_U^\dagger + \Pi_u(K - K_u - q_{u|\text{in}})\mathcal{U}_{u|\text{in}}\hat{b}_U \Big\} \,.
\end{aligned} \tag{105}
$$

Formula (19) is modified in a similar way:

$$
\begin{aligned}
\hat{\psi}_d(K) = \int_0^\infty \frac{\mathrm{d}\omega}{\sqrt{2\pi}} \Big\{ & \Pi_d(K - K_d - q_{d1|\text{out}})\mathcal{U}_{d1|\text{out}}(S_{d1,u}\hat{b}_U + S_{d1,d1}\hat{b}_{D1} + S_{d1,d2}\hat{b}_{D2}^\dagger) \\
& + \Pi_d(K - K_d + q_{d1|\text{out}})\mathcal{W}_{d1|\text{out}}^*(S_{d1,u}^*\hat{b}_U^\dagger + S_{d1,d1}^*\hat{b}_{D1}^\dagger + S_{d1,d2}^*\hat{b}_{D2}) \\
& + \Pi_d(K - K_d - q_{d2|\text{out}})\mathcal{U}_{d2|\text{out}}(S_{d2,u}\hat{b}_U + S_{d2,d1}\hat{b}_{D1} + S_{d2,d2}\hat{b}_{D2}^\dagger) \\
& + \Pi_d(K - K_d + q_{d2|\text{out}})\mathcal{W}_{d2|\text{out}}^*(S_{d2,u}^*\hat{b}_U^\dagger + S_{d2,d1}^*\hat{b}_{D1}^\dagger + S_{d2,d2}^*\hat{b}_{D2}) \\
& + \Pi_d(K - K_d + q_{d1|\text{in}})\mathcal{W}_{d1|\text{in}}^*\hat{b}_{D1}^\dagger + \Pi_d(K - K_d - q_{d1|\text{in}})\mathcal{U}_{d1|\text{in}}\hat{b}_{D1} \\
& + \Pi_d(K - K_d - q_{d2|\text{in}})\mathcal{U}_{d2|\text{in}}\hat{b}_{D2}^\dagger + \Pi_d(K - K_d + q_{d2|\text{in}})\mathcal{W}_{d2|\text{in}}^*\hat{b}_{D2} \Big\} \,.
\end{aligned} \tag{106}
$$

In the following we present the results for the flat profile configuration. In this case $K_u = K_d \equiv K_0$ and we note $k = K - K_0$, $q = Q - K_0$.

We first evaluate the one-body term $\langle \hat{N}(K) \rangle$ which has contributions coming from both $\langle \hat{\psi}_u^\dagger(K)\hat{\psi}_u(K) \rangle$ and $\langle \hat{\psi}_d^\dagger(K)\hat{\psi}_d(K) \rangle$. The double integral over $\omega$ and $\omega'$ defining these terms is reduced to a single integral by means of the contractions (52). In this integral one can safely discard overlap terms such as $\Pi_u(k - q_{u|\text{out}}(\omega))\Pi_u(k - q_{u|\text{in}}(\omega))$ when $\sigma_u \to \infty$ since $\Pi_u(K)$ is proportional to $\delta_u^{(1)}(K)$. One thus gets terms generically of the form of the one resulting from the contraction of the first term of the integral in the r.h.s. of (105) with its hermitian

conjugate, which reads:

$$\int_0^\infty \frac{d\omega}{2\pi} \, |S_{u,u}(\omega)\,\mathcal{U}_{u|\text{out}}(\omega)|^2 \, n_U(\omega) \, |\Pi_u(k - q_{u|\text{out}}(\omega))|^2 =$$
$$\int_{-\infty}^0 \frac{dp}{2\pi} \left| \frac{\partial \omega_{u|\text{out}}}{\partial p} \right| |S_{u,u}\,\mathcal{U}_{u|\text{out}}|^2 \, n_U \, |\Pi_u(k-p)|^2 = \frac{\sigma_u \Lambda_u^2}{\sqrt{8\pi}} \left| \frac{\partial \omega_{u|\text{out}}}{\partial k} \right| |S_{u,u}\,\mathcal{U}_{u|\text{out}}|^2 \, n_U \,. \tag{107}$$

In the integral of the second term of (107) one has made the change of variable $p = q_{u|\text{out}}(\omega)$ and all the $\omega$-dependent terms have to be evaluated at $\omega_{u|\text{out}}(p)$. In the last term of (107) one has used the fact that $|\Pi_u(k-p)|^2$ is proportional to $\delta_u^{(2)}(k-p)$ and all the $\omega$-dependent terms have to be evaluated at $\omega_{u|\text{out}}(k)$. As can be checked for instance by comparison with the similar contribution to $\langle \hat{N}(k) \rangle$ in Sec. 3.2, using the correct windowing for the Fourier transform, instead of the singular term $\delta(k-k)$ obtained with the schematic rules R1 and R2, one gets now a factor $\sigma_u \Lambda_u^2 / \sqrt{8\pi}$ for the terms issued from the upstream window and a factor $\sigma_d \Lambda_d^2 / \sqrt{8\pi}$ for the terms issued from the downstream window. As an illustration, the formulas equivalent to (37) and (38) (which, we recall, correspond to the adiabatic and zero temperature situation) read here

$$\langle \hat{N}(k < 0) \rangle = |S_{u,d2}|^2 \big|_{\omega_{u|\text{out}}(k)} \times \frac{\sigma_u \Lambda_u^2}{\sqrt{8\pi}} + \left( |S_{d2,u}|^2 + |S_{d2,d1}|^2 \right)_{\omega_{d2|\text{out}}(k)} \times \frac{\sigma_d \Lambda_d^2}{\sqrt{8\pi}}. \tag{108}$$

and

$$\langle \hat{N}(k > 0) \rangle = |S_{d1,d2}|^2 \big|_{\omega_{d1|\text{out}}(k)} \times \frac{\sigma_d \Lambda_d^2}{\sqrt{8\pi}} \,. \tag{109}$$

It now remains to evaluate the two-body function $G_2(K, Q)$. This involves four integrations over $\omega$, two of which disappear when using the contraction rules (52). In the contributions to $G_2(K, Q)$ one has to distinguish the diagonal terms — i.e., intra-channel correlations — and the crossed ones — inter-channel. The evaluation of the diagonal terms is simpler, and we only state the results: Instead of the singular term $\delta^2(k-q)$ (such as obtained for instance in the diagonal terms of (57) and (58)) one gets a term $\Lambda_u^4(\sigma_u/4\sqrt{\pi})\delta_u^{(1)}(k-q)$ for the contributions from the upstream windowing and a term $\Lambda_d^4(\sigma_d/4\sqrt{\pi})\delta_d^{(1)}(k-q)$ for the contributions from the downstream windowing.

One now has all the tools for determining the effect of the windowing on the evaluation of the intra-channel correlation signals, of the type $g_2(K, K)_{u|\text{out}}$ for instance:

$$g_2(K, K)_{u|\text{out}} = \frac{G_2(K, K)_{u|\text{out}}}{\langle \hat{N}(K) \rangle_{u|\text{out}}^2} + 1 \,, \tag{110}$$

where $\langle \hat{N}(K) \rangle_{u|\text{out}}$ and $G_2(K, K)_{u|\text{out}}$ are the $u|\text{out}$ contributions to $\langle \hat{N}(K) \rangle$ and to $G_2(K, K)$. With the correct rules presented above, one gets $g_2(K, K)_{u|\text{out}} = 2$, as in the main text. The same result holds true for all the intra-channel correlation terms.

We present the evaluation of the inter-channel terms in more detail, because it is less straightforward than the one of the diagonal terms, and also because it involves considerations relevant to the experimental detection scheme. Let us focus on the $u|\text{out}$-$d2|\text{out}$ contribution for instance. As done above [Eq. (107)] we illustrate the general case by studying one of the many contributions to $G_2|_{u|\text{out}-d2|\text{out}}$. In the four field quantity (30), one has a product of

four integrals of the type (105) and (106). For instance, one of the double contractions of terms issued from these integrals is

$$
\left[\mathcal{U}_{u|\text{out}}S_{u,d2}\right]^*_{\omega_1}\left[\mathcal{W}_{d2|\text{out}}S_{d2,d2}\right]_{\omega_2}\left[\mathcal{U}_{u|\text{out}}S_{u,d2}\right]_{\omega_3}\left[\mathcal{W}_{d2|\text{out}}S_{d2,d2}\right]^*_{\omega_4}\times
$$
$$
\langle\hat{b}_{D2}(\omega_1)\hat{b}^\dagger_{D2}(\omega_2)\hat{b}^\dagger_{D2}(\omega_3)\hat{b}_{D2}(\omega_4)\rangle\ . \tag{111}
$$

The contractions are evaluated using the finite temperature rules (52), and the contribution of the term corresponding to (111) can be written as the products of two independent integrals, $I$ and $J$:

$$
I(k,q) = \mathrm{e}^{\mathrm{i}(kX_u+qX_d)}\int_0^\infty\frac{\mathrm{d}\omega}{2\pi}(1+n_{D2})S_{d2,d2}S^*_{u,d2}\mathcal{U}^*_{u|\text{out}}\mathcal{W}_{d2|\text{out}}
$$
$$
\Pi_u(k-q_{u|\text{out}})\Pi_d(q+q_{d2|\text{out}})\mathrm{e}^{-\mathrm{i}q_{u|\text{out}}X_u}\mathrm{e}^{\mathrm{i}q_{d2|\text{out}}X_d}\ , \tag{112}
$$

$$
J(k,q) = \mathrm{e}^{-\mathrm{i}(kX_u+qX_d)}\int_0^\infty\frac{\mathrm{d}\omega}{2\pi}n_{D2}S^*_{d2,d2}S_{u,d2}\mathcal{U}_{u|\text{out}}\mathcal{W}^*_{d2|\text{out}}
$$
$$
\Pi_u(k-q_{u|\text{out}})\Pi_d(q+q_{d2|\text{out}})\mathrm{e}^{\mathrm{i}q_{u|\text{out}}X_u}\mathrm{e}^{-\mathrm{i}q_{d2|\text{out}}X_d}\ . \tag{113}
$$

The two integrals have similar forms. Each appears with a prefactor which disappears when the product $I\times J$ is performed: we thus drop this prefactor in the following and denote $\tilde{I}$ and $\tilde{J}$ the integrals where this prefactor is removed. We now focus on the evaluation of $\tilde{I}$; after a change of variable $p = q_{u|\text{out}}(\omega)$ it reads

$$
\tilde{I}(k,q) = \int_0^\infty\frac{\mathrm{d}p}{2\pi}A(\omega_{u|\text{out}}(p))\Pi_u(k-p)\Pi_d(q+q_{d2|\text{out}}(\omega_{u|\text{out}}(p)))\mathrm{e}^{-\mathrm{i}q_{u|\text{out}}X_u}\mathrm{e}^{\mathrm{i}q_{d2|\text{out}}(\omega_{u|\text{out}}(p))X_d}\ , \tag{114}
$$

where $A(\omega) = (1+n_{D2})S_{d2,d2}S^*_{u,d2}\mathcal{U}^*_{u|\text{out}}\mathcal{W}_{d2|\text{out}}|\partial\omega_{u|\text{out}}/\partial p|$.

For presenting the results it is easier to work in a simple regime where the dispersion relations are dispersionless; this will be assumed in the remaining of this appendix, the general result being given in the final formula (126). In this case $\partial\omega_{u|\text{out}}/\partial k = V_u - c_u \equiv V_{u|\text{out}}$ and $\partial\omega_{d2|\text{out}}/\partial k = V_d + c_d \equiv V_{d2|\text{out}}$ and[8] one can write $q_{d2|\text{out}}(\omega_{u|\text{out}}(p)) = -\gamma p$, where $\gamma \equiv -V_{u|\text{out}}/V_{d2|\text{out}} > 0$, cf. Fig. 10 (the notation $\gamma$ is temporarily introduced to make the computations easier to follow). Then (114) reads

$$
\tilde{I}(k,q) = \Lambda_u\Lambda_d\frac{\sigma_u\sigma_d}{2}\int_0^\infty\frac{\mathrm{d}p}{2\pi}A(\omega_{u|\text{out}}(p))\exp\{T(p,k,q)\} \tag{115}
$$

where

$$
T(p,k,q) = -\frac{\sigma_u^2}{4}(k-p)^2 - \frac{\sigma_d^2}{4}(q-\gamma p)^2 - \mathrm{i}p(X_u+\gamma X_d)
$$
$$
= -\frac{\sigma_u^2+\gamma^2\sigma_d^2}{4}(p-P(k,q))^2 - \frac{\sigma_u^2\sigma_d^2(\gamma k-q)^2}{4(\sigma_u^2+\gamma^2\sigma_d^2)} - Z(k,q)\ , \tag{116}
$$

with

$$
P(k,q) = \frac{\sigma_u^2 k + \sigma_d^2\gamma q - 2\mathrm{i}(X_u+\gamma X_d)}{\sigma_u^2+\gamma^2\sigma_d^2}\ , \tag{117}
$$

---

[8]We will use indifferently the notations $\partial\omega_\ell/\partial k$ or $V_\ell$ (with $\ell = u|\text{out}$ or $d2|\text{out}$) in the following of this appendix.

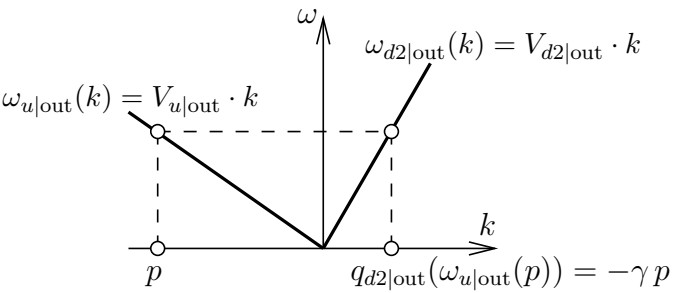

Figure 10: Dispersion relations in the long wavelength (dispersionless) limit. $V_{u|\text{out}} = V_u - c_u (< 0)$, $V_{d2|\text{out}} = V_d + c_d$ and $\gamma \equiv -V_{u|\text{out}}/V_{d2|\text{out}}$ $(> 0)$.

and

$$Z(k,q) = \frac{(X_u + \gamma X_d)^2}{\sigma_u^2 + \gamma^2 \sigma_d^2} + \mathrm{i}\, \frac{(X_u + \gamma X_d)(\sigma_u^2 k + \sigma_d^2 \gamma q)}{4(\sigma_u^2 + \gamma^2 \sigma_d^2)} \;. \tag{118}$$

It suffices to evaluate the integral (115) for $\sigma_u$ and $\sigma_d \to \infty$, which is the relevant limit as explained in the main text (cf. Sec. 2.4.1). In this case $A(\omega_{u|\text{out}}(p))$ is a weakly dependent function of $p$ compared with the rapidly varying exponent, and the integral in (115) can be computed by means of the steepest descent method. This amounts to evaluate $A(\omega_{u|\text{out}}(p))$ for $p = P(k,q)$ and to compute the remaining Gaussian integral. The result reads

$$\tilde{I}(k,q) = \Lambda_u \Lambda_d A(\omega_{u|\text{out}}(P)) V_{d2|\text{out}}\, \delta_{u.d}^{(3)}(V_{u|\text{out}} k + V_{d2|\text{out}} q)\, \exp\{-Z(k,q)\}\;, \tag{119}$$

where

$$\delta_{u.d}^{(3)}(K) = \sqrt{\frac{\sigma_u^2 \sigma_d^2/4\pi}{V_{d2|\text{out}}^2 \sigma_u^2 + V_{u|\text{out}}^2 \sigma_d^2}}\; \exp\left\{ \frac{-\sigma_u^2 \sigma_d^2\, K^2/4}{V_{d2|\text{out}}^2 \sigma_u^2 + V_{u|\text{out}}^2 \sigma_d^2}\right\}\;, \tag{120}$$

is again an approximation of the Dirac $\delta$-function. The term $\exp\{-Z\}$ in (119) induces a damping which is not present in the schematic approach presented in the main text. This term can be removed if one imposes $X_u = -\gamma X_d$, i.e.,

$$\frac{X_u}{V_{u|\text{out}}} = \frac{X_d}{V_{d2|\text{out}}}\;. \tag{121}$$

This relation has a simple physical interpretation: the time taken by an elementary excitation pertaining to the $u|$out channel to go from the horizon to the center ($X_u < 0$) of the upstream detection zone has to be the same as the time taken by its partner ($d2|$out channel) to go from the horizon to the center ($X_d > 0$) of the downstream detection zone. Note that this relation depends on the signal one is interested in (here the $u|\text{out} - d2|\text{out}$ channel): For other channels (say the $u|\text{out} - d1|\text{out}$ channel) the condition (121) will be modified and the centers of the window functions have to be shifted accordingly.

If the condition (121) is not fulfilled, the measured correlation will be damped compared to the perfect result [presented in the main text]. Note also that when this condition is fulfilled, since $q = \gamma k$ from the $\delta_{u.d}^{(3)}$ contribution in (119) one has $P(k,q) = k$ and, in this equation, $A(\omega)$ is evaluated at $\omega_{u|\text{out}}(k)$ as expected. Once condition (121) is realized, the product

$I \times J = \tilde{I} \times \tilde{J}$ is found to be equal to

$$I \times J = \Lambda_u^2 \Lambda_d^2 (1 + n_{D2}) n_{D2} \left| S_{d2,d2} S_{u,d2}^* \widetilde{\mathcal{U}}_{u|\text{out}} \widetilde{\mathcal{W}}_{d2|\text{out}} \right|^2 \left[ \delta_{u.d}^{(3)} (V_{u|\text{out}} k + V_{d2|\text{out}} q) \right]^2 . \qquad (122)$$

The same contribution evaluated with the less rigorous approach presented in the main text yields to a very similar expression, where the $\Lambda_u^2 \Lambda_d^2$ prefactor is missing and where the term $\left[ \delta_{u.d}^{(3)} (V_{u|\text{out}} k + V_{d2|\text{out}} q) \right]^2$ is replaced by $\delta^2(\omega_{u|\text{out}}(k) - \omega_{d2|\text{out}}(-q))$.

Finally, we consider the evaluation of the normalized inter-channel correlator

$$g_2(K, Q)_{u|\text{out}-d2|\text{out}} = \frac{G_2(K, Q)_{u|\text{out}-d2|\text{out}}}{\langle \hat{N}(K) \rangle_{u|\text{out}} \langle \hat{N}(Q) \rangle_{d2|\text{out}}} + 1 . \qquad (123)$$

When evaluating the fraction appearing in the r.h.s. of (123) along the line $\omega_{u|\text{out}}(k) = \omega_{d2|\text{out}}(-q)$ one obtains a ratio identical to the one obtained in Eq. (60) of the main text, multiplied by a factor

$$\left| \frac{\partial \omega_{u|\text{out}}}{\partial k} \cdot \frac{\partial \omega_{d2|\text{out}}}{\partial q} \right| \frac{\left[ \delta_{u.d}^{(3)}(0) \right]^2}{\Lambda_u^2 \sigma_u \Lambda_d^2 \sigma_d / 8\pi} = 2 \frac{\sigma_u \sigma_d |V_{u|\text{out}}| V_{d2|\text{out}}}{\sigma_u^2 V_{d2|\text{out}}^2 + \sigma_d^2 V_{u|\text{out}}^2} . \qquad (124)$$

This term is equal to unity, as it should, only if

$$\frac{\sigma_u}{|V_{u|\text{out}}|} = \frac{\sigma_d}{V_{d2|\text{out}}} , \qquad (125)$$

i.e., if the width of the window functions $\Pi_u(x)$ and $\Pi_d(x)$ are in the same ratio (121) as their center.

We recall that we have used a simplified linear dispersion relation for deriving the relations (121) and (125). However, there is dispersion in the system; this means that these relations have to be adapted for each $k$ and $q$ along a specific correlation line: they should read in the $u|\text{out} - d2|\text{out}$ case considered here

$$\frac{X_u}{\partial \omega_{u|\text{out}}/\partial k} = \frac{X_d}{\partial \omega_{d2|\text{out}}/\partial q} , \quad \text{and} \quad \frac{\sigma_u}{|\partial \omega_{u|\text{out}}/\partial k|} = \frac{\sigma_d}{\partial \omega_{d2|\text{out}}/\partial q} . \qquad (126)$$

The same condition has been already derived by de Nova, Sols and Zapata in Ref. [34].

# B  Non adiabatic effects (explicit results)

In this appendix we give the explicit form of the different contributions to the correlators (67), (68) and (69) discussed in subsection 3.3. The results are valid at finite temperature, and also *in situ*. The expressions are simplified as much as possible to ease readability:

$$\begin{aligned}
\text{Diag}_{<0} = \Big[ & |\alpha_u|^2 \Big( |S_{u,u}|^2 n_U + |S_{u,d1}|^2 n_{D1} + |S_{u,d2}|^2 (1 + n_{D2}) \Big) \\
& + |\alpha_d|^2 \Big( |S_{d2,u}|^2 (1 + n_U) + |S_{d2,d1}|^2 (1 + n_{D1}) + |S_{d2,d2}|^2 n_{D2} \Big) \\
& + |\alpha_d|^2 n_{D1} + |\alpha_d|^2 n_{D2} + |\beta_u|^2 (1 + n_U) \\
& + |\beta_d|^2 \Big( |S_{d1,u}|^2 (1 + n_U) + |S_{d1,d1}|^2 (1 + n_{D1}) + |S_{d1,d2}|^2 n_{D2} \Big) \Big]^2 ,
\end{aligned} \qquad (127)$$

$$O(|\alpha|^4)_{<0} = \tag{128}$$
$$|\alpha_u|^2|\alpha_d|^2\big|S_{u,u}^*S_{d2,u}n_U + S_{u,d1}^*S_{d2,d1}n_{D1} + S_{u,d2}^*S_{d2,d2}(1+n_{D2})\big|^2\delta^2(\omega_{u|\text{out}}(k) - \omega_{d2|\text{out}}(-q))$$
$$+|\alpha_u|^2|\alpha_d|^2|S_{u,d1}|^2n_{D1}^2\delta^2(\omega_{u|\text{out}}(k) - \omega_{d1|\text{in}}(q))$$
$$+|\alpha_u|^2|\alpha_d|^2|S_{u,d2}|^2n_{D2}(1+n_{D2})\delta^2(\omega_{u|\text{out}}(k) - \omega_{d2|\text{in}}(-q))$$
$$+|\alpha_d|^4|S_{d2,d1}|^2n_{D1}(1+n_{D1})\delta^2(\omega_{d2|\text{out}}(-k) - \omega_{d1|\text{in}}(q))$$
$$+|\alpha_d|^4|S_{d2,d2}|^2n_{D2}^2\delta^2(\omega_{d2|\text{out}}(-k) - \omega_{d2|\text{in}}(-q)) \ ,$$

$$O(|\alpha|^2|\beta|^2)_{<0} = \tag{129}$$
$$|\alpha_u|^2|\beta_u|^2|S_{u,u}|^2n_U(1+n_U)\delta^2(\omega_{u|\text{out}}(k) - \omega_{u|\text{in}}(-q))$$
$$+|\alpha_d|^2|\beta_u|^2|S_{d2,u}|^2(1+n_U)^2\delta^2(\omega_{d2|\text{out}}(-k) - \omega_{u|\text{in}}(-q))$$
$$+|\alpha_u|^2|\beta_d|^2|S_{u,u}^*S_{d_1u}n_U + S_{u,d1}^*S_{d1,d1}n_{D1} + S_{u,d2}^*S_{d1,d2}(1+n_{D2})|^2\delta^2(\omega_{u|\text{out}}(k) - \omega_{d1|\text{out}}(-q))$$
$$+|\alpha_d|^2|\beta_d|^2|S_{d2,u}S_{d1,u}^*n_U + S_{d2,d1}S_{d1,d1}^*n_{D1} + S_{d2,d2}S_{d1,d2}^*(1+n_{D2})|^2\delta^2(\omega_{d2|\text{out}}(-k) - \omega_{d1|\text{out}}(-q))$$
$$+|\alpha_d|^2|\beta_d|^2|S_{d1,d1}|^2n_{D1}(1+n_{D1})\delta^2(\omega_{d1|\text{in}}(k) - \omega_{d1|\text{out}}(-q))$$
$$+|\alpha_d|^2|\beta_d|^2|S_{d1,d2}|^2n_{D2}^2\delta^2(\omega_{d2|\text{in}}(-k) - \omega_{d1|\text{out}}(-q)) \ ,$$

$$O(|\beta|^4)_{<0} = |\beta_u|^2|\beta_d|^2|S_{d1,u}|^2(1+n_U)^2\delta^2(\omega_{u|\text{in}}(-k) - \omega_{d1|\text{out}}(-k)) \ , \tag{130}$$

$$\text{Diag}_{>0} = \Big[ \ |\alpha_d|^2\Big(|S_{d1,u}|^2n_U + |S_{d1,d1}|^2n_{D1} + |S_{d1,d2}|^2(1+n_{D2})\Big) \tag{131}$$
$$+|\alpha_u|^2n_U + |\beta_d|^2(1+n_{D1}) + |\beta_d|^2(1+n_{D2})$$
$$+|\beta_d|^2\Big(|S_{d2,u}|^2n_U + |S_{d2,d1}|^2n_{D1} + |S_{d2,d2}|^2(1+n_{D2})\Big)$$
$$+|\beta_u|^2\Big(|S_{u,u}|^2(1+n_U) + |S_{u,d1}|^2(1+n_{D1}) + |S_{u,d2}|^2n_{D2}\Big)\Big]^2 \ ,$$

$$O(|\alpha|^4)_{>0} = |\alpha_d|^2|\alpha_u|^2|S_{d1,u}|^2n_U^2\delta^2(\omega_{d1|\text{out}}(k) - \omega_{u|\text{in}}(q)) \ , \tag{132}$$

$$O(|\alpha|^2|\beta|^2)_{>0} = \tag{133}$$
$$|\alpha_d|^2|\beta_d|^2|S_{d1,u}^*S_{d2,u}n_U + S_{d1,d1}^*S_{d2,d1}n_{D1} + S_{d1,d2}^*S_{d2,d2}(1+n_{D2})|^2\delta^2(\omega_{d1|\text{out}}(k) - \omega_{d2|\text{out}}(q))$$
$$+|\alpha_d|^2|\beta_d|^2|S_{d1,d2}|^2(1+n_{D2})^2\delta^2(\omega_{d1|\text{out}}(k) - \omega_{d2|\text{in}}(q))$$
$$+|\beta_d|^2|\alpha_u|^2|S_{d2,u}|^2n_U^2\delta^2(\omega_{d2|\text{out}}(k) - \omega_{u|\text{in}}(q))$$
$$+|\alpha_d|^2|\beta_u|^2|S_{d1,u}^*S_{u,u}n_U + S_{d1,d1}^*S_{u,d1}n_{D1} + S_{d1,d2}^*S_{u,d2}(1+n_{D2})|^2\delta^2(\omega_{d1|\text{out}}(k) - \omega_{u|\text{out}}(-q))$$
$$+|\beta_u|^2|\alpha_u|^2|S_{u,u}|^2n_U(1+n_U)\delta^2(\omega_{u|\text{out}}(-k) - \omega_{u|\text{in}}(q))$$
$$+|\alpha_d|^2|\beta_d|^2|S_{d1,d1}|^2n_{D1}(1+n_{D1})\delta^2(\omega_{d1|\text{out}}(k) - \omega_{d1|\text{in}}(-q)) \ ,$$

$$O(|\beta|^4)_{>0} = \tag{134}$$
$$|\beta_d|^4|S_{d2,d1}|^2n_{D1}(1+n_{D1})\delta^2(\omega_{d2|\text{out}}(k) - \omega_{d1|\text{in}}(-q))$$
$$+|\beta_d|^4|S_{d2,d2}|^2(1+n_{D2})^2\delta^2(\omega_{d2|\text{out}}(k) - \omega_{d2|\text{in}}(q))$$

$$+|\beta_d|^2|\beta_u|^2|S_{d2,u}^*S_{u,u}n_U + S_{d2,d1}^*S_{u,d1}n_{D1} + S_{d2,d2}^*S_{u,d2}(1+n_{D2})|^2\delta^2(\omega_{d2|\text{out}}(k) - \omega_{u|\text{out}}(-q))$$
$$+|\beta_d|^2|\beta_u|^2|S_{u,d1}|^2n_{D1}^2\delta^2(\omega_{d1|\text{in}}(-k) - \omega_{u|\text{out}}(-q))$$
$$+|\beta_d|^2|\beta_u|^2|S_{u,d2}|^2(1+n_{D2})n_{D2}\delta^2(\omega_{d2|\text{in}}(k) - \omega_{u|\text{out}}(-q)) \ ,$$

$$
\begin{aligned}
A = \quad & Re\Big[\alpha_u\beta_u\Big(|S_{u,u}|^2(1+n_U) + |S_{u,d1}|^2(1+n_{D1}) + |S_{u,d2}|^2n_{D2}\Big) \\
& +\alpha_d\beta_d\Big(|S_{d2,u}|^2n_U + |S_{d2,d1}|^2n_{D1} + |S_{d2,d2}|^2(1+n_{D2})\Big) \\
& +\beta_d\alpha_d\Big(|S_{d1,u}|^2n_U + |S_{d1,d1}|^2n_{D1} + |S_{d1,d2}|^2(1+n_{D2})\Big) \\
& +\alpha_d\beta_d(1+n_{D1}) + \alpha_d\beta_d(1+n_{D2}) + \beta_u\alpha_u n_U\Big] \\
& \Big[\alpha_u\beta_u\Big(|S_{u,u}|^2n_U + |S_{u,d1}|^2n_{D1} + |S_{u,d2}|^2(1+n_{D2})\Big) \\
& +\alpha_d\beta_d\Big(|S_{d2,u}|^2(1+n_U) + |S_{d2,d1}|^2(1+n_{D1}) + |S_{d2,d2}|^2n_{D2}\Big) \\
& +\beta_d\alpha_d\Big(|S_{d1,u}|^2(1+n_U) + |S_{d1,d1}|^2(1+n_{D1}) + |S_{d1,d2}|^2n_{D2}\Big) \\
& +\alpha_d\beta_d n_{D1} + \alpha_d\beta_d n_{D2} + \beta_u\alpha_u(1+n_U)\Big] \ ,
\end{aligned}
\tag{135}
$$

$$O(|\alpha|^4) = \tag{136}$$
$$|\alpha_u|^2|\alpha_d|^2\Big|S_{u,u}^*S_{d1,u}n_U + S_{u,d1}^*S_{d1,d1}n_{D1} + S_{u,d2}^*S_{d1,d2}(1+n_{D2})\Big|^2\delta^2(\omega_{u|\text{out}}(k) - \omega_{d1|\text{out}}(q))$$
$$+|\alpha_d|^4\Big|S_{d2,u}^*S_{d1,u}n_U + S_{d2,d1}^*S_{d1,d1}n_{D1} + S_{d2,d2}^*S_{d1,d2}(1+n_{D2})\Big|^2\delta^2(\omega_{d2|\text{out}}(-k) - \omega_{d1|\text{out}}(q))$$
$$+|\alpha_u|^4|S_{u,u}|^2n_U^2\delta^2(\omega_{u|\text{out}}(k) - \omega_{u|\text{in}}(q))$$
$$+|\alpha_d|^4|S_{d1,d1}|^2n_{D1}^2\delta^2(\omega_{d1|\text{in}}(k) - \omega_{d1|\text{out}}(q))$$
$$+|\alpha_d|^2|\alpha_u|^2|S_{d2,u}|^2n_U(1+n_U)\delta^2(\omega_{d2|\text{out}}(-k) - \omega_{u|\text{in}}(q))$$
$$+|\alpha_d|^4|S_{d1,d2}|^2n_D(1+n_{D2})\delta^2(\omega_{d2|\text{in}}(-k) - \omega_{d1|\text{out}}(q)) \ ,$$

$$O(|\alpha|^2|\beta|^2) = \tag{137}$$
$$|\alpha_u|^2|\beta_d|^2|S_{u,u}^*S_{d2,u}n_U + S_{u,d1}^*S_{d2,d1}n_{D1} + S_{u,d2}^*S_{d2,d2}(1+n_{D2})|^2\delta^2(\omega_{u|\text{out}}(k) - \omega_{d2|\text{out}}(q))$$
$$+|\alpha_u|^2|\beta_d|^2|S_{u,d1}|^2n_{D1}(1+n_{D1})\delta^2(\omega_{u|\text{out}}(k) - \omega_{d1|\text{in}}(-q))$$
$$+|\alpha_u|^2|\beta_d|^2|S_{u,d2}|^2(1+n_{D2})^2\delta^2(\omega_{u|\text{out}}(k) - \omega_{d2|\text{in}}(q))$$
$$+|\alpha_d|^2|\beta_d|^2|S_{d2,d1}|^2(1+n_{D1})^2\delta^2(\omega_{d2|\text{out}}(-k) - \omega_{d1|\text{in}}(-q))$$
$$+|\alpha_d|^2|\beta_d|^2|S_{d2,d2}|^2n_{D2}(1+n_{D2})\delta^2(\omega_{d2|\text{out}}(-k) - \omega_{d2|\text{in}}(q))$$
$$+|\alpha_d|^2|\beta_u|^2|S_{d2,u}S_{u,u}^*n_U + S_{d2,d1}S_{u,d1}^*n_{D1} + S_{d2,d2}S_{u,d2}^*(1+n_{D2})|^2\delta^2(\omega_{d2|\text{out}}(-k) - \omega_{u|\text{out}}(-q))$$
$$+|\alpha_d|^2|\beta_d|^2|S_{d2,d1}|^2n_{D1}^2\delta^2(\omega_{d1|\text{in}}(k) - \omega_{d2|\text{out}}(q))$$
$$+|\alpha_d|^2|\beta_u|^2|S_{u,d1}|^2n_{D1}(1+n_{D1})\delta^2(\omega_{d1|\text{in}}(k) - \omega_{u|\text{out}}(-q))$$
$$+|\alpha_d|^2|\beta_d|^2|S_{d2,d2}|^2n_{D2}(1+n_{D2})\delta^2(\omega_{d2|\text{in}}(-k) - \omega_{d2|\text{out}}(q))$$
$$+|\alpha_d|^2|\beta_u|^2|S_{u,d2}|^2n_{D2}^2\delta^2(\omega_{d2|\text{in}}(-k) - \omega_{u|\text{out}}(-q))$$
$$+|\beta_u|^2|\alpha_d|^2|S_{d1,u}|^2n_U(1+n_U)\delta^2(\omega_{u|\text{in}}(-k) - \omega_{d1|\text{out}}(q))$$

$$+|\beta_d|^2|\alpha_u|^2|S_{d1,u}|^2 n_U(1+n_U)\delta^2(\omega_{d1|\text{out}}(-k)-\omega_{u|\text{in}}(q)) \,,$$

$$O(|\beta|^4)= \tag{138}$$
$$|\beta_d|^2|\beta_u|^2|S_{d2,u}|^2 n_U(1+n_U)\delta^2(\omega_{u|\text{in}}(-k)-\omega_{d2|\text{out}}(q))$$
$$+|\beta_u|^4|S_{u,u}|^2(1+n_U)^2\delta^2(\omega_{u|\text{in}}(-k)-\omega_{u|\text{out}}(-q))$$
$$+|\beta_d|^4|S_{d1,u}^*S_{d2,u}n_U+S_{d1,d1}^*S_{d2,d1}n_{D1}+S_{d1,d2}^*S_{d2,d2}(1+n_{D2})|^2\delta^2(\omega_{d1|\text{out}}(-k)-\omega_{d2|\text{out}}(q))$$
$$+|\beta_d|^4|S_{d1,d1}|^2(1+n_{D1})^2\delta^2(\omega_{d1|\text{out}}(-k)-\omega_{d1|\text{in}}(-q))$$
$$+|\beta_d|^4|S_{d1,d2}|^2 n_{D2}(1+n_{D2})\delta^2(\omega_{d1|\text{out}}(-k)-\omega_{d2|\text{in}}(q))$$
$$+|\beta_d|^2|\beta_u|^2|S_{d1,u}^*S_{u,u}n_U+S_{d1,d1}^*S_{u,d1}n_{D1}+S_{d1,d2}^*S_{u,d2}(1+n_{D2})|^2\delta^2(\omega_{d1|\text{out}}(-k)-\omega_{u|\text{out}}(-q)) \,.$$

## C  Subsonic flow in the presence of a localized obstacle

In this appendix we consider the scattering of a stationary subsonic flow onto a *localized external potential* and we assume that the downstream flow is also subsonic. This is a special case of the situation considered in section 4. The configuration is illustrated in Fig. 11. We show in this case that – when the non-linearity coefficient $g$ is $x$-independent – the far-upstream velocity and density are equal to the far-downstream velocity and density: $V_u = V_d$ and $n_u = n_d$.

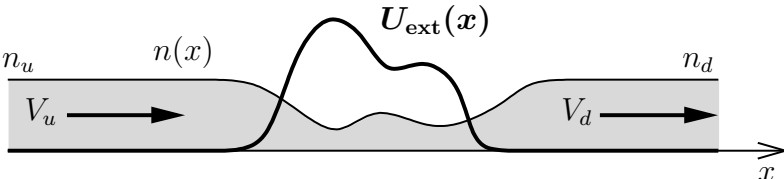

Figure 11: Sketch of the situation considered in the present appendix. The far upstream and far downstream asymptotic flows are both subsonic. The obstacle is represented by a localized external potential $U_{\text{ext}}(x)$.

Let's initially assume that the far upstream Mach number ($M_u = V_u/c_u$) and the far downstream one ($M_d = V_d/c_d$) are both less than unity, possibly different, and also that $V_u$ ($n_u$) may be different from $V_d$ ($n_d$). In our stationary setting, from the conservation of current and the definition of the speed of sound one gets

$$\frac{V_d}{V_u}=\frac{n_u}{n_d}=\left(\frac{c_u}{c_d}\right)^2 \equiv X \,, \tag{139}$$

where the last equality defines the quantity $X$.

The equality of the upstream and downstream chemical potentials reads

$$\tfrac{1}{2}mV_u^2+gn_u=\tfrac{1}{2}mV_d^2+gn_d \,. \tag{140}$$

Plugging (139) into (140) yields a third order equation for the quantity $X$. This equation can be cast under the form

$$(X - 1)(X^2 + X - 2\,M_u^{-2}) = 0 \ . \tag{141}$$

If, for the time being, one discards the trivial solution $X = 1$, the only other positive solution is $X = \frac{1}{2}(-1 + \sqrt{1 + 8M_u^{-2}}\,)$. Then, the far downstream Mach number is $M_d = V_d/c_d = M_u X^{3/2}$. It can easily be checked that for $M_u < 1$ (which has been assumed above) this expression yields for $M_d$ a value larger than 1: the downstream flow is supersonic, which contradicts our hypothesis. This means that the trivial solution $X = 1$ is the only acceptable one. From (139) one then gets the desired result: $n_u = n_d$ and $V_u = V_d$.

Note that the same result also holds true when the flow is supersonic both upstream and downstream: also in this case one has $n_u = n_d$ and $V_u = V_d$. An important outcome of this remark is that, in the presence of a localized obstacle, as soon as one is able to prove that the asymptotic upstream and downstream flow velocities are different, one can be sure that a sonic horizon has been realized.

Since for a localized obstacle the far upstream and far downstream characteristics of a subsonic flows are identical, the general formulas given in Sec. 4 simplify due to the following remarks:

(i) Since the far upstream and far downstream flows have the same density and velocity, $\omega_{u|\text{out}}(k) = \omega_{d|\text{in}}(k) = \omega(k < 0)$ and $\omega_{d|\text{out}}(k) = \omega_{u|\text{in}}(k) = \omega(k > 0)$.

(ii) $\mathcal{U}_\ell$ and $\mathcal{W}_\ell$ are functions of $\omega$ and $q_\ell(\omega)$ only (cf. their expression in [25]), whereas $\widetilde{\mathcal{U}}_\ell$ and $\widetilde{\mathcal{W}}_\ell$ are functions of $k$ and $\omega_\ell(k)$ only.

(iii) As a result of points (i) and (ii) above, in (89) all the $\widetilde{\mathcal{U}}_\ell$'s can be written as $\widetilde{\mathcal{U}}(k)$ and all the $\widetilde{\mathcal{W}}_\ell$'s can be written as $\widetilde{\mathcal{W}}(-k)$.

Note that these simplifications are only possible for a barrier of finite extent. If, for instance, one considers a flat profile configuration where the upstream and the downstream regions are both subsonic, the upstream and the downstream speeds of sound are not the same (because the upstream and downstream nonlinear coefficient are not the same) and point (i) above is not valid.

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
