# Peer review of "Momentum correlations as signature of sonic Hawking radiation in Bose-Einstein condensates"

_SciPost Physics, doi:SciPost Phys. 4, 019 (2018)_

## Round 1 · Referee Report · Anonymous (Referee 1) · 2018-2-19

Strengths

The manuscript is thorough and interesting.

Weaknesses

Several items are unclear, as mentioned in the "Requested changes" section.

Report

The manuscript is an in-depth study of the momentum correlations resulting from sonic Hawking radiation in a BEC. The authors gave a brief overview of the results in Ref. 17, but the manuscript is much more detailed than Ref. 17, and presents many additional results. The manuscript compares three different types of configurations, including the waterfall, the delta peak, and the flat profile. Furthermore, the effects of finite temperature and non-adiabaticity are studied. It is found that the correlation signal indicates non-separability even up to rather high temperatures, particularly in the waterfall configuration.

The manuscript is both interesting and thorough. I recommend its publication. However, I do have a few comments. I hope that the authors will consider making the changes necessary to answer the comments.

Figs. 6 and 7 show curves for a condensate which is in thermal equilibrium in the lab frame before the formation of the horizon. Is that possible? How would a condensate be formed with a thermal distribution which is centered around a non-zero momentum in the condensate frame?

The sentence at the bottom of p. 25 “We see from the results in Appendix B that the terms already present in the adiabatic regime are now multiplied by a factor alpha^4” seems unclear. Where does one see the terms which are already present in the adiabatic regime? Perhaps one example would be Eq. 59 for finite temperature. One should then compare this with Eq. 69 combined with Eq. 136. The sentence should be explained more thoroughly. Furthermore, it should be explicitly stated in Section 3.3 that Appendix B includes finite temperatures. This is in addition to the statement which appears in the appendix itself.

Also at the bottom of p. 25, it is stated that “This results in a very complicated pattern”. Is this pattern shown in the manuscript? Is it Fig. 8?

Requested changes

  1. Figs. 6 and 7 show curves for a condensate which is in thermal equilibrium in the lab frame before the formation of the horizon. Is that possible? How would a condensate be formed with a thermal distribution which is centered around a non-zero momentum in the condensate frame?

  2. The sentence at the bottom of p. 25 “We see from the results in Appendix B that the terms already present in the adiabatic regime are now multiplied by a factor alpha^4” seems unclear. Where does one see the terms which are already present in the adiabatic regime? Perhaps one example would be Eq. 59 for finite temperature. One should then compare this with Eq. 69 combined with Eq. 136. The sentence should be explained more thoroughly. Furthermore, it should be explicitly stated in Section 3.3 that Appendix B includes finite temperatures. This is in addition to the statement which appears in the appendix itself.

  3. Also at the bottom of p. 25, it is stated that “This results in a very complicated pattern”. Is this pattern shown in the manuscript? Is it Fig. 8?

  • validity: top
  • significance: high
  • originality: good
  • clarity: high
  • formatting: excellent
  • grammar: excellent

Author:  Nicolas Pavloff  on 2018-03-15  [id 230]

(in reply to Report 1 on 2018-02-19)

---

## Round 1 · Referee Report · Anonymous (Referee 2) · 2018-2-21

Strengths

  • clarity of writing
  • accessiblity of technically complicated subject
  • motivation for the need of these results

Weaknesses

  • none big enough to list here

Report

Sonic Hawking radiation is the paired emission of quantum sound-waves from a location in a flowing Bose-Einstein condensate where the flow speed of the latter makes
a transition from subsonic to supersonic (sonic horizon). The mathematical origin of this emission is identical to that of Hawking radiation emitted from the event horizon of a black-hole,
for intuitively the same reasons: The sound wave propagates with the locally varying speed of sound, hence the sonic horizon representes a point of no return for it, in the same way an event horizon of a black hole
represents a point of no return for any particle propagating within the GR space-time.
The analogy extends to rigorous mathematics as the wave-equations for phonons in a BEC can be shown to be equivalent to the Klein-Gordon equation of a scalar particle in a curved space time, with metric tensor defined by
density and velocity fields of the BEC.
Experiments have recently reported first observations of this analogue Hawking effect in a BEC, while this will not be possible around common size astrophysical black holes.
Experiments are challenging however, and a promising route to obtain clearer data are momentum correlations (rather than currently used density correlations.

The present manuscript provides a large host of useful technical details for the theoretical and practical dealings with those momentum correlations, thus I can recommend its publication.
The results will provide crucial guidance for experiments to try to obtain the best possible signals.
The article is also very well written, given the technicality of the subject. I would only suggest that the minor points listed under "requested changes" are addressed.

Finally I have to apologize to the authors and the editor for the disproportionally long time it has taken to produce this report.

Requested changes

  • It seems the manuscript is substantially extending the detail level on Ref [17] where the idea to study sonic black holes via momentum correlations was first introduced by a larger team including the same authors. I find the paragraph in the introduction relating to this (2nd on page 3) a little too minimal: Here the authors should clearly state what was already discussed in [17] and what is newly provided here or elsewhere. This will make an article of this size much more accessible in the future.

  • The authors cite a variety of their own work as first references for a "\delta peak" configuration. Here PRA 76 013608 (2007) which specifically addresses a very similar setup in the analogue gravity context should also be cited. This also discussed the possibility of a transverse Thomas-Fermi profile as discussed in section 5.

  • I suggest that the captions of Fig. 4/5 could attempt a 1-2 line definition of what the correlation-lines are, duplicating the more rigorous definition in the text to make these plots more accessible the reader.

  • validity: top
  • significance: good
  • originality: good
  • clarity: top
  • formatting: excellent
  • grammar: excellent

Author:  Nicolas Pavloff  on 2018-03-15  [id 231]

(in reply to Report 2 on 2018-02-21)

---

## Round 2 · List of Changes

Besides the point already discussed in the reply to the referee's comments we added some references, corrected typos and added a new subsection label for clarity

---

## Editorial Decision

published